# LEARNING DIVERSE SKILLS FOR BEHAVIOR MODELS WITH MIXTURE OF EXPERTS

## ABSTRACT

Imitation learning has demonstrated strong performance in robotic manipulation by learning from large-scale human demonstrations. While existing models excel at single-task learning, it is observed in practical applications that their performance degrades in the multi-task setting, where interference across tasks leads to an averaging effect. To address this issue, we propose to learn diverse skills for behavior models with Mixture of Experts, referred to as Di-BM. Di-BM associates each expert with a distinct observation distribution, enabling experts to specialize in sub-regions of the observation space. Specifically, we employ energy-based models to represent expert-specific observation distributions and jointly train them alongside the corresponding action models. Our approach is plug-and-play and can be seamlessly integrated into standard imitation learning methods. Extensive experiments on multiple real-world robotic manipulation tasks demonstrate that Di-BM significantly outperforms state-of-the-art baselines. Moreover, fine-tuning the pretrained Di-BM on novel tasks exhibits superior data efficiency and the reusable of expert-learned knowledge.

## 1 INTRODUCTION

Learning versatile and generalist robotic policies is crucial for developing intelligent and practically applicable robots. Recent advances in large-scale datasets have significantly boosted the capability of robotic manipulation. Although strong performance across diverse real-world manipulation tasks has been demonstrated by recent works, training behavior models on large-scale multi-task robotic datasets remains challenging due to multi-modal action distribution and versatile skills including in complicated tasks.

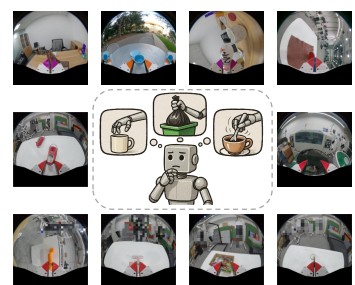

Figure 1: Complex robotic tasks are hypothesized to be decomposed into a set of primitive skills, which are mastered by different experts.

Diffusion Policy (Chi et al., 2023) was the first to incorporate diffusion models (Ho et al., 2020) to address the problem of multi-modal action distribution, and this idea has since been adopted by several follow-up works (Black et al., 2024; Liu et al., 2024; Barreiros et al., 2025). However, learning versatile skills remains a major challenge, as training a single model across multiple tasks leads to interference between different tasks and results in degraded performance. To improve the performance of multi-task learning, we consider that complex robotic tasks can be decomposed into a set of primitive skills, as illustrated in Figure 1, and adopt the Mixture of Experts (MoE) mechanism (Cai et al., 2025).

MoE has shown remarkable success in Large Language Models (LLMs) (Lepikhin et al., 2020; Du et al., 2022; Fedus et al., 2022) but remains underexplored in robotics. In LLMs, MoE is primarily used to expand model parameters while maintaining computational sparsity, without explicitly modeling the specific distributions each expert excels at. In this paper, each expert is encouraged to specialize in a subset of primitive skills, while the overall skill domain is collaboratively covered by the ensemble of experts. A naïve approach would be to manually partition the dataset into different skill categories and assign them to experts (Yang et al., 2025); however, this incurs substantial

labeling costs and relies heavily on human intuition, making accurate categorization difficult. To enable experts to automatically acquire primitive skills, we draw inspiration from self-paced diverse skill learning (Celik et al., 2022; 2024). Concretely, we employ an Energy-Based Model (EBM) to characterize the observation distribution of each expert, measuring how strongly an observation is favored by a given expert. Meanwhile, a parameterized gating network models these distributions and determines which observations are allocated to which experts. Each expert is trained on the subset of observations it favors, while the gating network is jointly optimized to identify the favored region of each expert and to ensure collectively coverage of the observation space.

In summary, we propose **Di-BM** - **Di**verse skill learning for **B**ehavior **M**odels designed to acquire primitive skills from multi-task datasets. By leveraging EBM-based observation distributions together with a gating network, Di-BM automatically allocates experts to its preferred domain. The main contributions of this paper are as follows:

- **Enhanced multi-task performance.** Extensive experiments on diverse real-world robotic manipulation tasks demonstrate that Di-BM achieves substantial improvements over state-of-the-art baselines in mult-task learning.
- **Diverse skill specialization.** Visualizations of expert-selection probabilities during deployment verify that experts indeed specialize in distinct domains.
- **Data-efficient fine-tuning.** Fine-tuning the pre-trained Di-BM on novel tasks yields superior performance with less training data compared to baselines, highlighting both its data efficiency and the reusability of expert-learned knowledge.

## 2 RELATED WORK

### 2.1 MIXTURE-OF-EXPERTS RELATED WORKS

Mixture-of-Experts (MoE) architectures (Jacobs et al., 1991; Jordan & Jacobs, 1994) were originally proposed to introduce a gating mechanism that dynamically routes inputs to a subset of specialized experts. This sparse activation has become a mainstream approach for scaling large language models (LLMs) (Fedus et al., 2022), enabling systems such as Switch Transformers (Fedus et al., 2022), DeepSeekMoE (Dai et al., 2024), and Mixtral-8x7B (Jiang et al., 2024) demonstrating improved task specialization and representational capacity while maintaining inference efficiency.

Beyond language modeling, MoE has also been explored in reinforcement learning (RL) policies (Celik et al., 2022; 2024), where experts parameterize distinct policy distributions and a gating network adaptively combines them. These approaches address task heterogeneity, improve multimodal policy parameterization, and mitigate gradient interference in multi-task learning. Some works (Li et al., 2023; Blessing et al., 2023) explore MoE in imitation learning by assigning a closed-form, per-sample weight to each expert within a batch. In contrast, our method employs a learnable gating network to estimate expert-selection probabilities. We hypothesize that this learnable gating mechanism better captures expert affinity and specialization in the observation space.

In robotics, MoE has shown promise for enhancing multi-task learning by modularizing policies into specialized skill experts. For instance, MoE has been applied to multitask locomotion (Huang et al., 2025; Song et al., 2024), where experts specialize in primitive motor skills such as walking, running, or jumping, as well as to multi-modal fusion settings (Yu et al., 2025; Yang et al., 2025), where experts process diverse sensor modalities. For robot manipulation tasks, MENTOR (Huang et al., 2024) introduces MoE and a task-oriented perturbation mechanism for visual reinforcement learning. SDP (Wang et al., 2024) uses a separate router for each task while sharing experts across tasks, whereas MoDE (Reuss et al., 2024) conditions the MoE router on the diffusion noise level. Distinct from the above methods, our approach leverages EBM-based observation distributions together with a gating network to automatically allocate experts to their preferred domains.

### 2.2 BEHAVIOR MODELS FOR ROBOT MANIPULATION

Behavior cloning, the most widely used approach in robot learning, enables task-specific robotic capabilities by imitating human demonstrations (Reuss et al., 2023) (Chi et al., 2023; Zhao et al.,

2023; 2024). Prior works (Lin et al., 2024; Chi et al., 2024) have shown that scaling the number of demonstrations improves generalization to novel environments and objects within a single task. Recently, Vision-Language-Action (VLA) models have emerged as a central paradigm for robotic learning, grounding visual and linguistic inputs into executable actions by leveraging large-scale, diverse datasets. Early efforts such as RT-2 (Zitkovich et al., 2023) reformulated robot control as a vision-language modeling problem by outputting actions as text tokens, while subsequent systems, including OpenVLA (Kim et al., 2024), RDT-1B (Liu et al., 2024), and $\pi_0$ (Black et al., 2024) demonstrated the benefits of introducing large-scale pretraining and multi-modal fusion for robotic manipulation.

Building on these developments, we extend the behavior cloning paradigm with an MoE framework, hypothesizing that complex robotic tasks can be decomposed into a set of primitive skills, each handled by specialized experts. With sparse activation, our approach enhances scalability and adaptability of behavior models to diverse tasks while adding minimal computational overhead.

## 3 Preliminary

### 3.1 MoE Architecture

The key idea of **Mixture of Experts (MoE)** (Lepikhin et al., 2020; Du et al., 2022; Fedus et al., 2022) is to route each input to the most suitable expert subnetworks via a gating mechanism, facilitating specialization and enabling more efficient computation. A standard MoE layer consists of $K$ expert networks, denoted as $\{E_e\}_{e=0}^{K-1}$, together with a gating network (or router) $\pi(\cdot|x)$, where $x$ denotes the input. In typical sparse implementations of MoE, the gating network $\pi(\cdot|x)$ predicts the probability of each expert to select a small subset of $k$ experts (commonly $k = 1$ or 2, with $k \ll K$) from the full pool of $K$ experts. The input is then routed to these $k$ chosen experts, whose outputs, $E_e(x)$, are subsequently aggregated, usually via a weighted sum with weights $\pi(e|x)$ provided by the gating network. Formally, the output of the MoE layer is expressed as $y(x) = \sum_{e \in TopK(\pi(\cdot|x))} \pi(e|x) E_e(x)$, where $TopK(\pi(\cdot|x))$ denotes the indices of the top-$k$ experts selected for input $x$.

While our method builds upon the MoE framework, several key modifications are introduced. First, to encourage each expert to focus on a distinct region of the observation space, we set $k = 1$, such that each input is routed to a single expert. Second, instead of assigning a separate gating network to each MoE layer, we employ a single shared router, which predicts probability from the observation input, across all MoE layers to accommodate the proposed diverse skill learning algorithm.

### 3.2 Self-Paced Diverse Skill Learning with MoE

This paper focuses on the imitation learning framework that is widely adopted in robot manipulation training, and the optimization objective is to minimize the training loss:

$$\max_{\pi(a|o)} \mathbb{E}_{p(o)}[\mathbb{E}_{\pi(a|o)}[-\mathcal{L}(a, \hat{a})]], \tag{1}$$

where $o$ denotes the observation, $p(o)$ the observation distribution induced by the environment, $\pi(a|o)$ the learned action policy, $a$ the ground-truth actions associated with $o$ in the dataset, and $\hat{a} \sim \pi(a|o)$ the predicted actions.

Self-paced diverse skill learning (Celik et al., 2022; 2024) incorporates Mixture-of-Experts (MoE) and learns expert-specific observation distributions. This allows each expert to specialize on a subset of observations, while collectively covering the entire observation space. It is worth noting that prior works mainly investigated multi-skill learning under reinforcement learning framework, whereas we focus on the case of imitation learning. Consequently, the notation differs slightly. In particular, the MoE policy can be formulated as $\pi(a|o) = \sum_e \pi(e|o)\pi(a|o, e)$, where $\pi(e|o)$ is the probability of selecting expert $e$ given observation $o$, and $\pi(a|o, e)$ denotes the action policy of expert $e$. Applying Bayes' rule, $\pi(e|o) = \pi(o|e)\pi(e)/\pi(o)$, and the policy can be rewritten as $\pi(a|o) = \sum_e \frac{\pi(o|e)\pi(e)}{\pi(o)}\pi(a|o, e)$ with $\pi(o) = \sum_e \pi(o|e)\pi(e)$. Here, $\pi(o|e)$ represents the observation distribution associated with expert $e$, which can be optimized to capture the observations

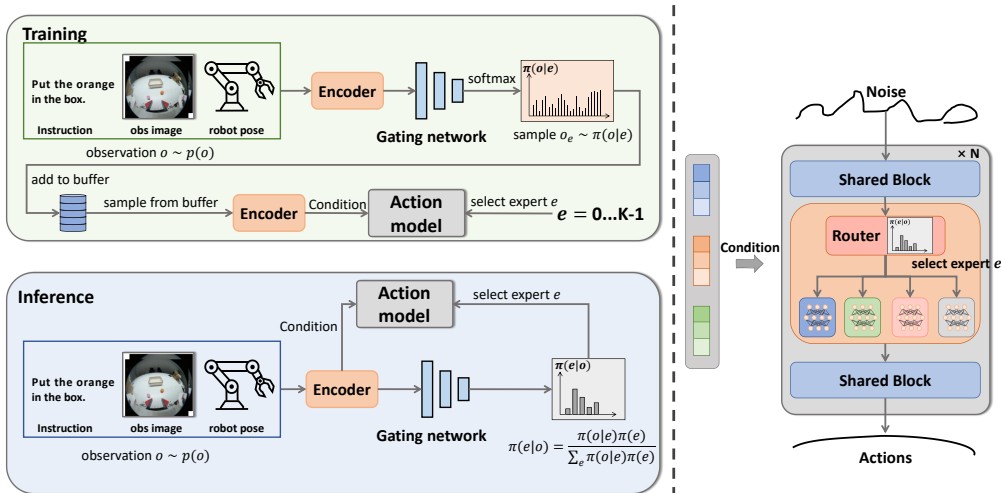

(a) The overall of Di-BM framework.                (b) Details of the action model.

Figure 2: The overall policy consists of an encoder, a gating network, and an action model. During training, the gating network assigns appropriate data to each expert for specialization, whereas during inference it selects the most suitable expert to handle the current observation.

preferred by the particular expert. The prior $\pi(e)$ is set to be uniform throughout this work, following Celik et al. (2024).

To explicitly optimize the policy, we adopt KL-regularization in the training objective. It is worth noting that Celik et al. (2022; 2024) introduce an additional entropy bonus term $H(\pi(a|o))$ here to encourage action diversity. In contrast, since diffusion models used in this paper inherently capture multi-modal distributions (Chi et al., 2023), we omit this term in our formulation.

$$\max_{\pi(a|o),\pi(o)} \mathbb{E}_{p(o)}[\mathbb{E}_{\pi(a|o)}[-\mathcal{L}(a,\hat{a})]] - \beta KL(p(o)||\pi(o)). \tag{2}$$

The KL term encourages the learned observation distribution $\pi(o)$ to match the environment distribution $p(o)$. Following Celik et al. (2022; 2024), the updates for $\pi(a|o,e)$ and $\pi(o|e)$ can be derived separately as:

$$\max_{\pi(a|o,e)} \mathbb{E}_{\pi(o|e),\pi(a|o,e)}[-\mathcal{L}(a,\hat{a})], \tag{3}$$

$$\max_{\pi(o|e)} \mathbb{E}_{\pi(o|e)}[\mathbb{E}_{\pi(a|o,e)}[-\mathcal{L}(a,\hat{a})] + \beta \log \tilde{\pi}(e|o)] + \beta H(\pi(o|e)), \tag{4}$$

where $\tilde{\pi}(e|o) = \pi_{old}(e|o)$ (no gradient) and $H(\pi(o|e)) = -\int_o \pi(o|e) \log \pi(o|e) do$ is the entropy bonus. Details of the equations are in the Appendix A.1.

## 4 APPROACH

### 4.1 PROBLEM FORMULATION AND OVERVIEW OF DI-BM

Di-BM is an autonomous diverse-skill robotic policy for manipulation, which inputs an visual observations, language instruction, and the current robot pose (collectively referred to as the observation), and outputs the corresponding action. As shown in Figure 2, the overall policy consists of three components: an encoder, a gating network, and an action model. The encoder extracts features from the observation, which are then used as the conditioning input for both the gating network and the action model. The gating network, in turn, produces outputs the observation distribution of each expert and computes the probability of selecting an expert (detailed in Section 4.2). The action model, composed of $K$ experts, then generates actions conditioned on the selected expert $e$ and the encoded observation features.

To better illustrate how these components interact in practice, we next describe the training and inference procedures of Di-BM. During training, the gating network assigns observations to the experts that favor them, and each expert is updated accordingly. At inference time, the gating network selects the most appropriate expert to process the given observation. The optimization procedure will be detailed in Section 4.3.

We adopt the Diffusion Policy (Chi et al., 2023) as our action model, which has demonstrated strong ability to capture multi-modal action distributions. MoE layers are interleaved between the shared blocks and the model architecture is detailed in Appendix A.2. We formulate the action model as Denoising Diffusion Probabilistic Models (DDPMs)(Ho et al., 2020):

$$\hat{a}^{k-1} = Denoise\big(a^k, f_\theta(a^k, o, k, e), k\big), \tag{5}$$

where $k$ denotes the diffusion step and $a^k$ is the noisy action at step $k$. Consequently, the loss function $\mathcal{L}$ in (3) and (4) takes the form of diffusion loss

$$\mathcal{L} = MSE\big(\epsilon^k, f_\theta(a^k, o, k, e)\big). \tag{6}$$

## 4.2 ENERGY-BASED MODEL FOR GATING NETWORK

We denote $\pi(o|e)$ as the observation distribution associated with expert $e$, characterizing how strongly an expert favors a particular observation $o$. To model these distributions, we adopt an Energy-Based Model (EBM), owing to its ability to capture sharp discontinuities and represent multi-modal distributions in complex environments (Florence et al., 2022). Following Celik et al. (2024), we parameterize each expert distribution $\pi(o|e)$ as:

$$\pi(o|e) = \exp(g_\phi(o, e))/Z_e, \tag{7}$$

where $g_\phi$ is the learnable gating network parameterized with $\phi$, and $Z_e = \int_{\mathbf{o}} \exp(g_\phi(\mathbf{o}, e))d\mathbf{o}$ is the normalizing constant for expert $e$. In practice, the normalizing constant $Z_e$ can be approximated by Monte Carlo estimation, i.e., $Z_e \approx \sum_{i=0}^{N-1} \exp(g_\phi(o_i, e))$, where $o_i \sim p(o)$ is sampled from the environment. To ensure that the EBM is exposed to critical regions of the observation space, we resample sufficiently large batches of observations $o \sim p(o)$ at each training iteration.

During training, $\pi(o|e)$ is used to sample the observations favored by each expert $e$, based on the output of the gating network. At inference time, $\pi(e|o) = \pi(o|e)\pi(e)/\pi(o)$ is employed to select the most specialized expert for a given observation, where $\pi(o) = \sum_e \pi(o|e)\pi(e)$. In this way, the gating network serves as the router in the MoE framework, dynamically assigning observations to the most appropriate expert.

## 4.3 TRAINING AND INFERENCE OF DI-BM

### 4.3.1 TRAINING

We update each expert $\pi(a|o, e)$ and its corresponding per-expert observation distribution $\pi(o|e)$ by maximizing the objectives defined in (3) and in (4), respectively. These decomposed objectives allow us to independently update $\pi(a|o, e)$ and $\pi(o|e)$, and to learn diverse skills for each expert. The overall training pipeline is summarized in Algorithm 1.

**Expert Update.** The expert policy $\pi(a|o, e)$ is updated following (3), which can be rewritten as:

$$\max_{f_\theta} \sum_{o_i \sim \pi(o|e)} [-MSE(\epsilon_i^k, \hat{\epsilon}_i^k)], \tag{8}$$

where $\pi(o|e) = \mathrm{softmax}(g_\phi(o, e))$ and $\hat{\epsilon}_i^k = f_\theta(a_i^k, o_i, k, e)$. At each update step for expert $e$, we sample $S$ pairs of $(o, a)$ from the training set according to $\pi(o|e)$ and add them to the buffer $\mathcal{B}$. As $o$ and $a$ occur in pairs in the dataset, we omit $a$ and denote sampling only by $o$ for notational simplicity. A minibatch $B'$ is then drawn from $\mathcal{B}$ (Algorithm 1). These samples are regarded as favorable to expert $e$, and we optimize $f_\theta$ by minimizing the diffusion loss over this selected subset.

**Per-expert Observation Distribution Update.** We update the observation distribution $\pi(o|e)$ according to (4), and the overall optimization objective can then be reformulated as

$$\max_{g_\phi} \sum_{o_i \sim p(o)} \pi(o_i|e)(-MSE(\epsilon_i^k, \hat{\epsilon}_i^k) + \beta \log \tilde{\pi}(e|o_i) - \beta \log \pi(o_i|e)), \tag{9}$$

where $\pi(o_i|e) = \text{softmax}(g_\phi(o_i, e))$. At each update step of the observation distribution, we sample a batch of $(o, a)$ pairs from the training set according to the environment distribution $p(o)$, denoted as $o_B$ in Algorithm 1. The second term encourages $\pi(o|e)$ to assign higher probability to regions of the observation space that are less covered by other experts, thereby reducing overlap. The third term acts as an entropy bonus, promoting broader coverage of the entire observation space.

**Training Loss.** In the prior work of Celik et al. (2024), the experts are updated sequentially, and the optimization of the expert network and the gating network is separated. However, we found that applying this sequential and separate update scheme led to severe training instability in our context, which involves significantly larger network parameters and parameter sharing among multiple experts. Consequently, we updates $f_\theta$ and $g_\phi$ jointly across all experts to ensure stable learning. The resulting training loss is defined as

$$
\begin{aligned}
Loss = &\sum_e \sum_{o_i \sim \pi(o|e)} MSE(\epsilon_i^k, \hat{\epsilon}_i^k) \\
&+ \gamma \sum_e \sum_{o_i \sim p(o)} \pi(o_i|e)\Big( \tilde{MSE}(\epsilon_i^k, \hat{\epsilon}_i^k) - \beta \log \tilde{\pi}(e|o_i) + \beta \log \pi(o_i|e) \Big),
\end{aligned}
\tag{10}
$$

where $\tilde{MSE}(\cdot|\cdot)$ and $\tilde{\pi}(e|o_i)$ denote that the gradient of them is stopped during backpropagation.

---

**Algorithm 1** Di-BM Training

---

**Input:** $N$ (max iterations), $K$ (number of experts), $S$ (samples per expert), $B$ (batch size), $B'$ (batch size for training experts), $T$ (training timesteps)
**Output:** $f_\theta, g_\phi$
    Initialize buffer $\mathcal{B} \leftarrow \emptyset$
    **for** $i = 0$ **to** $N - 1$ **do**
        Sample $B$ observations $o_B \sim p(o)$ from training dataset
        **for** $e = 0$ **to** $K - 1$ **do**
            $\pi(o_B|e) = \text{softmax}(g_\phi(o_B, e))$
            Sample $S$ observations $o_S \sim \pi(o_B|e)$
            $o_S \rightarrow \mathcal{B}$
            Get $B'$ observations $o_{B'} \leftarrow \mathcal{B}$
            $k \sim Uniform(\{0, ..., T-1\})$
            $\hat{\epsilon}_{B'} = f_\theta(a_{B'}, o_{B'}, k, e)$
            $\hat{\epsilon}_B = f_\theta(a_B, o_B, k, e)$;          // `torch.no_grad`
        Update $f_\theta$ and $g_\phi$ with (10)

---

### 4.3.2 INFERENCE

During inference, the observation distribution $\pi(o|e)$ is employed to determine which expert should be selected, as illustrated in Algorithm 2. Since $\pi(o|e)$ is parameterized as energy-based model, the normalizing constant $Z_e$ is required. In practice, we compute $Z_e = \sum_{o_i \sim p(o)} \exp(g_\phi(o, e))$ from the training dataset and reuse it directly during deployment.

---

**Algorithm 2** Di-BM Inference

---

**Input:** $o_t$ (observation at time $t$), $f_\theta, g_\phi, Z, T$ (inference timesteps)
**Output:** $a_t$ (action)
    $\pi(o_t|e) = \exp(g_\phi(o_t, e))/Z_e$
    $e_t \sim \pi(e|o_t)$, where $\pi(e|o_t) = \pi(o_t|e)\pi(e)/\sum_e \pi(o_t|e)\pi(e)$
    $\hat{a}_t^T \sim \mathcal{N}(\mathbf{0}, \mathbf{I})$
    **for** $k = T$ **to** $1$ **do**
        $\hat{a}_t^{k-1} = Denoise(\hat{a}_t^k, f_\theta(\hat{a}_t^k, o_t, k, e), k)$
    $a_t = \hat{a}_t^0$

---

## 5 Simulation Experiments

### 5.1 Implementation Details

For simulation evaluation, we evaluate our method on the RoboTwin 2.0 (Chen et al., 2025) benchmark. Specifically, we collect a multi-task dataset consisting of 50 trajectories for each of the 8 tasks to train the policy. As our focus is on enabling the behavior model to acquire diverse skills from multi-task data, we train a single policy across all 8 tasks and evaluate its performance on this same task set for 100 trials per task.

We incorporate our proposed Diverse Skill Learning mechanism into the CNN-based variant of Diffusion Policy (Chi et al., 2023). This involves replacing one-fourth of the convolutional layers in the UNet architecture with Mixture-of-Experts (MoE) layers. We compare our method with the following strong baselines: 1) standard Diffusion Policy (DP), 2) SDP (Wang et al., 2024), which uses a separate router for each task while sharing experts across tasks, and 3) MoDE (Reuss et al., 2024), which uses the noise level as an input to the router. All MoE-based baselines (SDP, MoDE, and our method) are implemented with the same CNN-based Diffusion Policy backbone and 5 experts in the MoE layers.

### 5.2 Simulation multi-task learning results

Table 1 summarizes the multi-task learning evaluation results on the RoboTwin 2.0 benchmark. Overall, our proposed method significantly outperforms all established baselines, especially in tasks that demand high manipulation precision (e.g., Beat block hammer, Handover block, etc.), which emphatically demonstrates the effectiveness and specialized capability of our multi-skill learning mechanism in the multi-task setting.

Table 1: Simulation performance in RoboTwin 2.0.

| Method | Adjust bottle | Beat block hammer | Click alarmclock | Dump bin bigbin | Grab roller | Handover block | Handover mic | Lift pot | Total |
|--------|---------------|-------------------|------------------|-----------------|-------------|----------------|--------------|----------|-------|
| DP | 0.82 | 0.06 | 0.85 | 0.55 | **0.95** | 0.31 | 0.52 | 0.83 | 0.61 |
| SDP | **0.95** | 0.15 | 0.73 | 0.35 | 0.79 | 0.05 | 0.42 | 0.24 | 0.46 |
| MoDE | 0.82 | 0.15 | 0.73 | 0.59 | 0.89 | 0.33 | 0.56 | 0.68 | 0.59 |
| Di-BM | 0.83 | **0.41** | **0.88** | **0.60** | 0.92 | **0.72** | **0.78** | **0.91** | **0.76** |

In our experiments, we observe that the coefficient $\beta$ of the KL regularization term in (2) has a significant impact on model performance. Therefore, we conduct an ablation study on $\beta$. Table 2 reports the results under different $\beta$ values. The results show that $\beta = 0.01$ yields the best performance. A large $\beta$ over-constrains the experts, preventing them from specializing in their respective domains, which leads to behavior similar to a standard diffusion policy. In contrast, a very small $\beta$ allows experts to "slack off" on the harder parts of the task, resulting in degraded performance. Further analysis is provided in Section 6.5.

Table 2: Ablation for $\beta$.

| $\beta$ | Avg. Success. |
|---------|---------------|
| 0.001 | 0.23 |
| 0.01 | **0.76** |
| 0.1 | 0.59 |
| 1 | 0.60 |
| 10 | 0.63 |

## 6 Real-world Experiments

### 6.1 Training Dataset

To collect sufficient data for multi-task manipulation, we adopt the Universal Manipulation Interface (UMI) (Chi et al., 2024), a user-friendly and low-cost hand-held gripper system for data collection. Using UMI, we independently gathered extensive demonstrations and further incorporated data from Lin et al. (2024), resulting in a total of $\sim 24$ hours of multi-task demonstrations. The dataset spans a wide range of manipulation tasks, including pouring drink, folding towel, etc. The details of these tasks are presented in Appendix A.3. We train a single policy on an aggregated dataset containing all 9 tasks and evaluate it on this same task set.

## 6.2 IMPLEMENTATION DETAILS

To demonstrate the effectiveness of our approach, we incorporate the diverse skill learning mechanism into Diffusion Policy (Chi et al., 2023) as the action model, considering both CNN-based and transformer-based variants. Notably, our method is plug-and-play and can be seamlessly integrated into existing methods. For the CNN-based version, we replace one-fourth of the convolutional layers in the UNet with MoE layers, while for the transformer-based version, MoE layers are incorporated into each feed-forward network (FFN). For feature extraction from visual and language observations, we adopt the CLS token from a pretrained CLIP Vision Transformer backbone, specifically ViT-B/16 (Radford et al., 2021). The model is trained for 70 epochs with the AdamW optimizer (Loshchilov & Hutter, 2017), using a batch size of 128 on eight NVIDIA H100 GPUs. A complete list of hyperparameters is provided in Appendix A.2. We use the Nova5 robot arm for our real-world evaluations. The model runs on an RTX 4090 with an inference time of 45 ms, which supports real-time execution.

## 6.3 REAL-WORLD MULTI-TASK LEARNING RESULTS

We evaluate our method on 9 real-world manipulation tasks, conducting 10 trials per task for both CNN-based and transformer-based variants. The details of these tasks are presented in Appendix A.3. As summarized in Table 3, our proposed diverse skill learning mechanism substantially improves behavior model performance under multi-task training. It is worth noting that training on multi-task datasets often suffers from the averaging effect, where tasks with fewer demonstrations are overshadowed by those with larger data volumes. Despite this challenge, our method achieves consistently strong performance across all 9 tasks.

Table 3: Real-world performance. "-C" and "-T" correspond to the CNN-based and transformer-based variants respectively. We categorize the tasks into "easy" and "hard" based on their operational precision requirements and complexity.

| Method | easy | | | | | | hard | | | Total |
|---|---|---|---|---|---|---|---|---|---|---|
| | Throw into trash | Open drawer | Close drawer | Fold towel (horizontally) | Push objects | Pick and place into basket | Pour drink | Rearrange cup | Stir in cup | |
| DP-C | 0.40 | **1.00** | 0.40 | 0.30 | 0.30 | 0.90 | 0.90 | 0.10 | 0.40 | 0.52 |
| DP-C-Di (5 experts) | **0.60** | 0.90 | **0.80** | **1.00** | **0.80** | **1.00** | **1.00** | **0.80** | **0.80** | **0.86** |
| DP-T | 0.50 | 0.50 | 0.00 | 0.40 | 0.10 | 0.40 | 0.60 | 0.10 | 0.50 | 0.34 |
| DP-T-Di (3 experts) | **0.90** | **0.90** | **0.50** | **0.60** | **0.70** | **0.90** | **0.80** | **0.20** | **0.80** | **0.70** |

## 6.4 ABLATION

**MoE Ablation.** To ensure that the observed performance improvements are not simply attributed to an increase in model capacity, we compare our method against two alternative MoE implementations in the DiffusionPolicy-C variant. The first, referred to as vanilla MoE, is the approach commonly adopted in LLMs (Dai et al., 2024), where a gating mechanism is introduced in each MoE layer. In this baseline, we incorporate vanilla MoE which employs a load-balancing loss (Fedus et al., 2022) to encourage uniform expert utilization and directly trains on the entire dataset. The second baseline, referred to as task-wise MoE, assigns experts based on task categories, using the manual partitioning strategy(Yang et al., 2025), with the specific task assignments. These two MoE baselines are detailed in Appendix A.4. The results, summarized in Table 4, demonstrate that our proposed approach, where experts autonomously acquire specialized skills, outperforms both baselines.

Table 4: Performance for different MoE formulation. We categorize the tasks into "easy" and "hard" based on their operational precision requirements and complexity.

| Method | easy | | | | | | hard | | | Total |
|---|---|---|---|---|---|---|---|---|---|---|
| | Throw into trash | Open drawer | Close drawer | Fold towel (horizontally) | Push objects | Pick and place into basket | Pour drink | Rearrange cup | Stir in cup | |
| Vanilla MoE | 0.40 | 0.70 | **0.90** | 0.50 | 0.40 | 0.90 | 0.80 | 0.10 | 0.50 | 0.58 |
| Task-wise MoE | **0.90** | 0.70 | 0.60 | 0.70 | 0.60 | **1.00** | 0.80 | 0.30 | 0.70 | 0.70 |
| Di-BM | 0.60 | **0.90** | 0.80 | **1.00** | **0.80** | **1.00** | **1.00** | **0.80** | **0.80** | **0.86** |

**Expert Ablation.** We evaluate the performance of individual expert networks in the Di-BM (DiffusionPolicy-C variant) with five experts, as shown in Table 5. The results indicate that although a single expert is capable of accomplishing an entire task, its performance is inferior to that of the combined experts. This suggests that in our approach, each expert is more proficient in specific phases of a task while being less effective in others. We further explore the performance of Di-BM with different numbers of experts, as detailed in Appendix A.6.

Table 5: Performance for individual expert.

|  | Expert 0 | Expert 1 | Expert 2 | Expert 3 | Expert 4 | Di-BM |
|---|---|---|---|---|---|---|
| Rearrange cup | 0.50 | 0.40 | 0.40 | 0.70 | 0.50 | **0.80** |
| Fold towel (horizontally) | 0.50 | 0.80 | 0.40 | 0.80 | 0.80 | **1.00** |
| Push objects | 0.70 | 0.60 | 0.60 | 0.60 | 0.70 | **0.80** |
| Total | 0.57 | 0.60 | 0.47 | 0.70 | 0.67 | **0.87** |

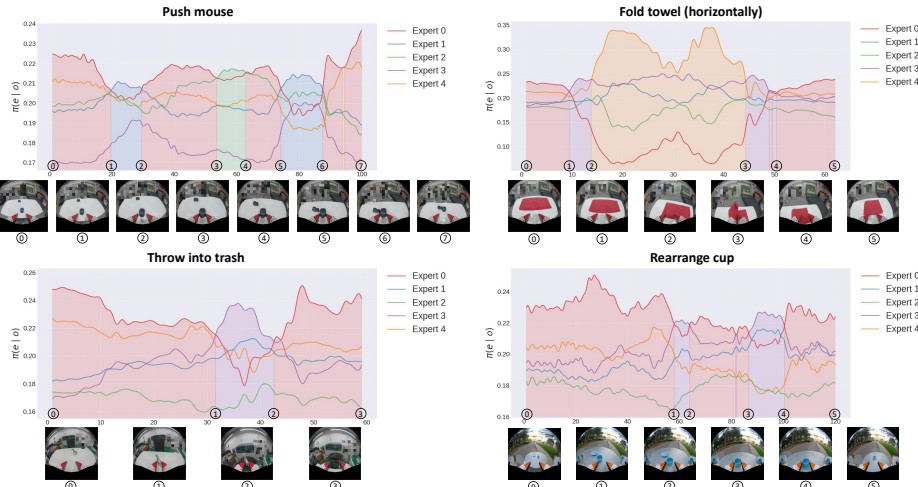

Figure 3: Visualization of $\pi(e|o)$ across different tasks, where the shaded regions indicate the currently dominant expert.

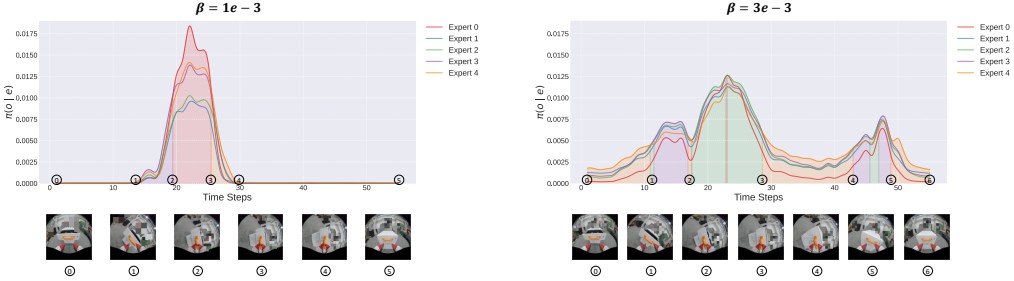

Figure 4: $\pi(o|e)$ under different $\beta$ settings. Here, $\pi(o|e) = \exp(g_\phi(o,e))/Z_e$, where $Z_e = \sum_{i=0}^{N-1} \exp(g_\phi(o,i))$ is estimated from 100 random samples in the dataset. A smaller $\beta$ leads all experts to "slack off".

## 6.5 ANALYSIS OF DIVERSE SKILL LEARNING

We visualize $\pi(e|o)$ for all experts in Figure 3, which illustrates which expert dominates different phases of each task. The results reveal that experts are not simply assigned to a single fixed task; instead, each expert becomes responsible for particular phases across multiple tasks. Notably, although we did not provide any explicit supervision to enforce the learning of primitive skills during training, the experts still developed distinct specializations. For example, Expert 0 dominates the

gripper translation behavior at the beginning of tasks and when retracting after completion, while Expert 1 specializes in alignment behaviors in the push mouse task, such as aligning the gripper with the mouse and the mouse with the camera. These findings indicate that our approach enables the model to autonomously achieve primitive skills in an unsupervised manner. We further analyze the learned condition feature corresponding to different experts with Di-BM with t-SNE (see Appendix A.5 for details).

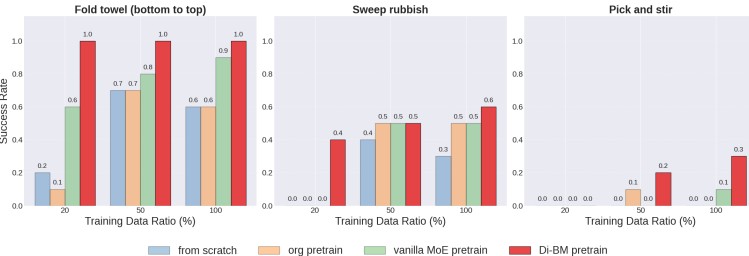

Figure 5: Performance on novel tasks with different training data ratios. "from scratch" is trained only on new data, "org pretrain" is fine-tuned from the pretrained original DiffusionPolicy, "vanilla MoE pretrain" is fine-tuned from the pretrained vanilla MoE variant, and "Di-BM pretrain" is fine-tuned from our pretrained model.

Moreover, while exploring the hyperparameter $\beta$, we find that the KL regularization term in (2) is crucial and indispensable. When $\beta$ is set too small, all experts tend to "slack off". As shown in Figure 4, we plot the curves of $\pi(o|e)$ on the same task segment with models trained under different values of $\beta$. With $\beta = 1\text{e-}3$, the gating network assigns very low probabilities (nearly zero) to actions that are hard to learn, causing all experts to avoid learning these challenging parts—a behavior we do not desire. In contrast, when $\beta$ is larger (e.g., $\beta = 3\text{e-}3$), this issue disappears.

### 6.6 ADVANTAGE IN POST-TRAINING

Building on the hypothesis that each expert in Di-BM specializes in a subset of primitive skills, we expect the pretrained Di-BM to adapt more efficiently to unseen tasks through post-training. To validate this hypothesis, we evaluate the model on three novel tasks—"fold towel (from bottom to top)", "sweep rubbish" and "pick and stir"—which are not included in the pretraining dataset (see Appendix A.3 for details). For each task, we collect 300 demonstrations and fine-tune the model using different ratios of this data. As shown in Figure 5, the pretrained Di-BM consistently achieves superior performance with significantly less data, demonstrating its strong data efficiency in adapting to new tasks.

## 7 CONCLUSION

In this work, we propose Di-BM, a Mixture of Experts (MoE) framework that enhances multi-task learning through autonomous specialization. An energy-based model characterizes the observation distribution of each expert, enabling data to be routed to the most suitable expert and thereby improving specialization. Experiments on real-world tasks demonstrate the effectiveness of Di-BM, while visualizations of $\pi(e|o)$ show that experts dynamically dominate different phases in long-horizon tasks. This autonomous MoE paradigm outperforms conventional MoE strategies in robotics, and fine-tuning the pretrained Di-BM on novel tasks further yields superior data efficiency.

Our approach still has several limitations that we aim to address in future work. First, our method has primarily been validated within the Diffusion Policy framework to demonstrate its benefits in multi-task learning. However, existing VLA models are typically built upon larger LLM backbones and trained on much larger-scale datasets. A promising direction is to incorporate the Di-BM learning paradigm into large-scale VLAs to further enhance manipulation skill learning. Second, we observe that the model is relatively sensitive to hyperparameters, particularly the coefficient $\beta$. Setting $\beta$ too high prevents experts from focusing on specific skills, while too small a value causes all experts to "slack off." Future work could explore adaptive strategies that dynamically adjust such parameters during training to improve the stability of multi-skill learning.

## ETHICS STATEMENT

This research does not involve human subjects, sensitive personal data, or practices related to dataset release. The experiments are conducted in controlled environments and do not raise privacy, security, or legal compliance concerns. We are not aware of any conflicts of interest or risks of discrimination, bias, or fairness issues. The work adheres to principles of research integrity and is intended solely for advancing the field of robotic learning.

## REPRODUCIBILITY STATEMENT

We have made every effort to ensure the reproducibility of our results. The implementation details of our method and experiments are provided in main text and appendix. We have included our code in the supplementary materials for reproducibility. Upon acceptance of the paper, the code will be released publicly to facilitate further research.

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

# A APPENDIX

## A.1 DERIVATION OF THE OPTIMIZATION OBJECTIVE

The total optimization objective is defined as

$$\max_{\pi(a|o),\pi(o)} \mathbb{E}_{p(o)}[\mathbb{E}_{\pi(a|o)}[-\mathcal{L}(a,\hat{a})]] - \beta KL(p(o)||\pi(o)). \tag{11}$$

Given the conditions

$$\pi(a|o) = \sum_e \pi(e|o)\pi(a|o,e), \tag{12}$$

$$\pi(o) = \sum_e \pi(o|e)\pi(e), \tag{13}$$

$$\log \pi(o) = \log \frac{\pi(o|e)\pi(e)}{\pi(e|o)}, \tag{14}$$

$$KL(p(o)||\pi(o)) = \int_o \pi(o) \log \frac{\pi(o)}{p(o)} do, \tag{15}$$

the objective can be reformulated as

$$J = \sum_e \pi(e)\mathbb{E}_{\pi(o|e)}[\mathbb{E}_{\pi(a|o,e)}[-\mathcal{L}(a,\hat{a})]] - \beta KL(p(o)||\pi(o)). \tag{16}$$

Expanding the KL term yields

$$J = \sum_e \pi(e)\mathbb{E}_{\pi(o|e)}[\mathbb{E}_{\pi(a|o,e)}[-\mathcal{L}(a,\hat{a})]] + \sum_e \pi(e)\mathbb{E}_{\pi(o|e)}[-\beta \log \pi(o|e) + \beta \log \pi(e|o) + \beta p(o)] + \beta H(e). \tag{17}$$

By assuming $p(o)$ and $\pi(e)$ to be uniform, the constant terms $p(o)$ and $H(e)$ can be omitted, leaving

$$J = \sum_e \pi(e)\mathbb{E}_{\pi(o|e)}[\mathbb{E}_{\pi(a|o,e)}[-\mathcal{L}(a,\hat{a})]] + \sum_e \pi(e)\mathbb{E}_{\pi(o|e)}[-\beta \log \pi(o|e) + \beta \log \pi(e|o)]. \tag{18}$$

**Expert Update.** By extracting the terms containing $\pi(a|o,e)$, the update objective for expert $e$ is

$$\max_{\pi(a|o,e)} \mathbb{E}_{\pi(o|e),\pi(a|o,e)}[-\mathcal{L}(a,\hat{a})]. \tag{19}$$

**Per-expert Observation Distribution Update.** By extracting the terms involving $\pi(o|e)$, the update objective for the observation distribution of expert $e$ becomes

$$\max_{\pi(o|e)} \mathbb{E}_{\pi(o|e)}[\mathbb{E}_{\pi(a|o,e)}[-\mathcal{L}(a,\hat{a})] + \beta \log \tilde{\pi}(e|o)] + \beta H(\pi(o|e)), \tag{20}$$

where $\tilde{\pi}(e|o) = \pi_{old}(e|o)$ is introduced following Celik et al. (2022) to optimize individual expert. The outer expectation $\mathbb{E}_{\pi(o|e)}$ is approximated via observation sampling, as in (9).

## A.2 MODEL DETAILS AND HYPERPARAMETERS

The instruction and observation images are separately encoded by CLIP (Radford et al., 2021) and subsequently fused via Feature-wise Linear Modulation (FiLM) (Perez et al., 2018). The resulting visual-language feature is then concatenated with the robot pose to form a global condition, which serves both as the input to the gating network and as the conditioning signal for the action model.

We list the hyperparameters for our CNN-based and transformer-based Di-BM in Table 6.

| Hyperparameter | DiffusionPolicy-C-Di | DiffusionPolicy-T-Di |
|---|---|---|
| Training denoising steps | 50 | 50 |
| Inference denoising steps | 16 | 16 |
| Number of experts | 5 | 3 |
| Samples per expert $S$ | 32 | 64 |
| Batch size $B$ | 128 | 256 |
| Batch size for training experts $B'$ | 32 | 256 |
| $\beta$ | 3e-3 | 3e-3 |
| $\gamma$ | 100 | 100 |

Table 6: Hyperparameters for DiffusionPolicy-C-Di and DiffusionPolicy-T-Di.

## A.3 TEST TASK VISUALIZATION

We test all pretrained models trained on pretraining dataset on 9 real-world tasks—"rearrange cup", "pour drink", "fold towel (horizontally)", "push objects", "stir in cup", "pick and place into basket", "throw into trash", "open drawer" and "close drawer", which will be described in Figure 6 and Table 7.

We test all fine-tuned models on 3 novel real-world tasks—"fold towel (from bottom to top)", "sweep rubbish" and "pick and stir", which will be described in Figure 7 and Table 8.

For each task, we test with objects of various colors and categories to increase the diversity of evaluation. All methods are evaluated under the same condition. For example, for the "push objects" task, each method is tested with each of "square box" * 2, "orange" * 2, "mouse" * 2, "cola" * 2 and "toy" * 2.

| Task | Description | Prompt |
|------|-------------|--------|
| Rearrange cup | Rotate the handle of the cup to the left, then pick up the cup and place it on the saucer. | Rearrange the espresso cup on the saucer with the handle facing left. |
| Pour drink | Grasp the drinking bottle, tilt it to pour the drink into a cup, and then place the bottle back on the coaster. | Pick up the drinking bottle, and pour water into a mug. |
| Fold towel (horizontally) | Grasp the left side of the towel and fold it toward the right / Grasp the right side of the towel and fold it toward the left. | Fold the yellow towel from left to right / right to left. Fold the white tablecloth from left to right. Fold the purple tablecloth from left to right. Fold the red tablecloth from left to right. Fold the blue tablecloth from right to left. Fold the green tablecloth from right to left. Fold the white tablecloth from right to left. Fold the yellow towel from right to left. Fold the red tablecloth from right to left. Fold the kleinblue tablecloth from right to left. Fold the brown towel from left to right. Fold the white tablecloth from right to left. Fold the green tablecloth from right to left. |
| Push objects | Gently grasp the front of objects and push them across the table to the designated position. | Push the square box to the appropriate location. Push the apple to the appropriate location. Push the peach to the appropriate location. Push the coaster to the appropriate location. Push the remote control to the appropriate location. Push the tea bottle to the appropriate location. Push the white canvas to the appropriate location. Push the red canvas to the appropriate location. Push the yellow canvas to the appropriate location. Push the basket to the appropriate location. Push the mouse to the appropriate location. Push the cola to the appropriate location. Push the orange to the appropriate location. Push the toy to the appropriate location. Push the black canvas to the appropriate location. Push the plastic box to the appropriate location. Push the battery to the appropriate location. Push the cube to the appropriate location. Push the green slippers to the appropriate location. Push the red tape to the appropriate location. |
| Stir in cup | Grasp the stirring stick inside the cup and stir the contents. | Stir inside the black cup to mix the contents. Stir inside the red cup to mix the contents. Stir inside the grey cup to mix the contents. Stir inside the green cup to mix the contents. Stir inside the yellow cup to mix the contents. Stir inside the white cup to mix the contents. Stir inside the blue cup to mix the contents. Stir inside the coffee cup to mix the contents. Stir inside the brown cup to mix the contents. Stir inside the enamel mug to mix the contents. Stir inside the plastic cup to mix the contents. Stir inside the steel cup to mix the contents. Stir inside the tea cup to mix the contents. |
| Pick and place into basket | Pick up a toy and place it into the basket. | Pick up the toy, and place the toy into the basket. Pick up the apple, and place the apple into the basket. Pick up the peach, and place the peach into the basket. Pick up the remote control, and place the remote control into the basket. Pick up the pink square, and place the pink square into the basket. Pick up the red tape, and place the red tape into the basket. Pick up the orange, and place the orange into the basket. |
| Throw into trash | Grasp an object and throw it into the trash. | Throw the apple and orange into the trash. Throw the banana and peach into the trash. Throw the vegetable crackers and cookies into the trash. Throw the cola and tea bottle into the trash. Throw the toy into the trash. |
| Open drawer | Grasp the drawer handle and pull it backward to open. | Grip the handle to open the drawer. |
| Close drawer | Grasp the drawer handle and push it forward to close. | Grip the handle to close the drawer. |

Table 7: Descriptions of real-world tasks with multiple prompts per task.

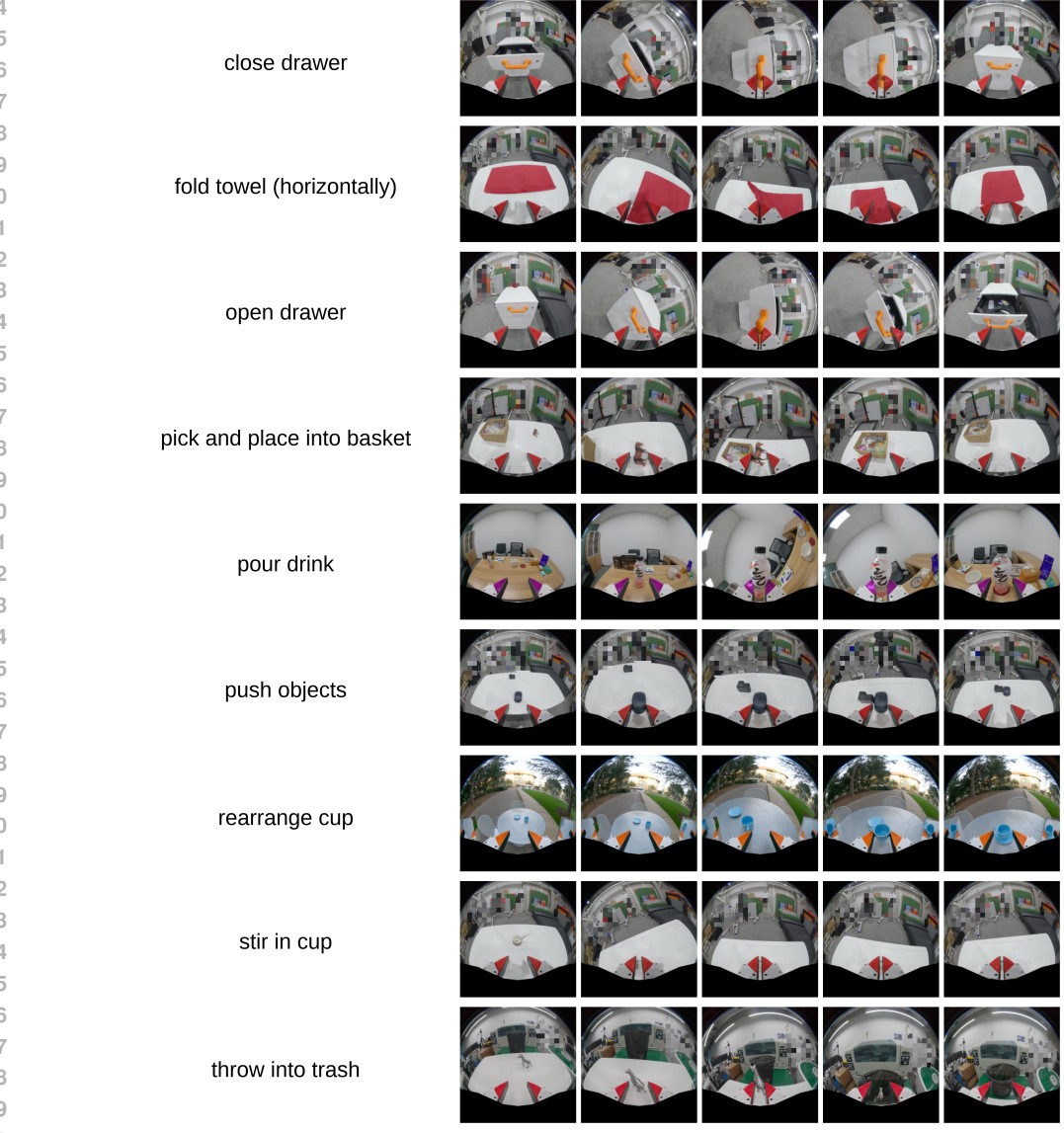

Figure 6: Visualization of 9 real-world tasks, including "rearrange cup", "pour drink", "fold towel (horizontally)", "push objects", "stir in cup", "pick and place into basket", "throw into trash", "open drawer" and "close drawer".

| Task | Description | Prompt |
|---|---|---|
| Fold towel (from bottom to top) | Grasp the bottom side of the towel and fold it toward the top. | Fold the brown towel from bottom to top. Fold the yellow towel from bottom to top. Fold the red towel from bottom to top. Fold the blue towel from bottom to top. Fold the light blue towel from bottom to top. |
| Sweep rubbish | Grasp the handle of the broom and sweep the rubbish on the table into the dustpan. | Sweep the rubbish from left to right. Sweep the rubbish from right to left. |
| Pick and stir | Grasp the straw placed in the black pen holder, pick it up, place it into the white cup, and stir. | Pick up the white straw and stir inside the white enamel cup with it. Pick up black straw and stir inside the white enamel cup with it. |

Table 8: Descriptions of 3 novel real-world tasks for post-training.

Figure 7: Visualization of 3 novel real-world tasks for post-training, including "fold towel (from bottom to top)", "sweep rubbish" and "pick and stir".

### A.4 BASELINE DETAILS

For the vanilla MoE baseline, we use 5 experts and adopt a gating network in each MoE layer. Following Dai et al. (2024), we employ a load-balancing loss to encourage uniform expert utilization, formulated as

$$\mathcal{L}_{\text{balance}} = N \cdot \sum_{i=1}^{N} f_i \cdot p_i \tag{21}$$

where

$$f_i = \frac{1}{B} \sum_{b=1}^{B} \mathbf{1}[s_b = i], \qquad \text{(fraction of tokens actually routed to expert } i)$$

$$p_i = \frac{1}{B} \sum_{b=1}^{B} g_i(x_b), \qquad \text{(average gating probability assigned to expert } i)$$

For the task-wise MoE baseline, we use 6 experts and manually assign them to specific task groups, as summarized in Table 9.

| Expert ID | Assigned Tasks |
|-----------|----------------|
| 0 | Pour drink |
| 1 | Open & Close drawer |
| 2 | Fold towel (horizontally) & Rearrange cup |
| 3 | Throw into trash |
| 4 | Stir in cup |
| 5 | Pick and place into basket & Push objects |

Table 9: Task-wise MoE baseline: expert-to-task assignment strategy.

## A.5 ANALYSIS OF CONDITION FEATURE FOR EXPERTS

We analyze the learned condition feature from the encoder with Di-BM. Specifically, we visualize the global condition feature using t-SNE, as shown in Figure 8. The conditional feature distributions corresponding to different experts exhibit a certain degree of separation, with each color representing one expert. Each expert is concentrated in a specific subregion of the feature space, indicating that the corresponding expert specializes in that region. This specialization effect is induced by the $\tilde{\pi}(e|o_i)$ term in (9).

At the same time, partial overlaps are also observed between experts, which aligns with our expectations. This is encouraged by the entropy bonus term $\pi(o_i|e)$ in (9). Moreover, since primitive skills are not strictly disjoint, the resulting feature distributions inevitably overlap. Thus, the observed distributions reflect a desirable balance: experts specialize in distinct regions of the feature space while collectively covering the entire feature space.

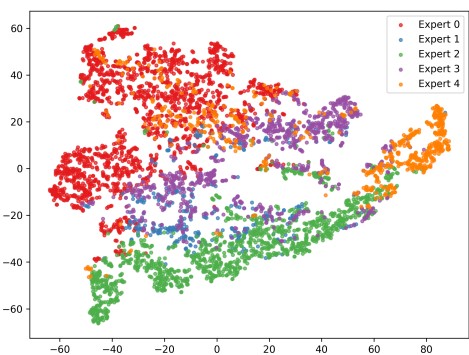

Figure 8: t-SNE visualization of condition feature.

## A.6 EXPLORATION OF EXPERT NUMBER

We further explore the effect of the number of experts by testing models with 3, 5, and 8 experts, as shown in Table 10 and Figure 9. The results indicate that increasing the number of experts to 8 does not lead to further performance improvements. We attribute this to the limited diversity of skills in our dataset, where 5 experts are already sufficient to cover the available primitive skills. Interestingly, in the 8-expert model, only 4–5 experts are actively utilized, while the others consistently receive low probabilities of $\pi(e|o)$. This suggests that the model naturally prunes underutilized experts when they provide little additional benefit. We hypothesize that with a larger and more diverse multi-task dataset, a scaling law with respect to the number of experts may emerge. Although increasing the coefficient $\beta$ of the KL regularization term could potentially encourage more balanced expert utilization, we did not pursue this direction further, as the 5-expert model already achieves strong performance.

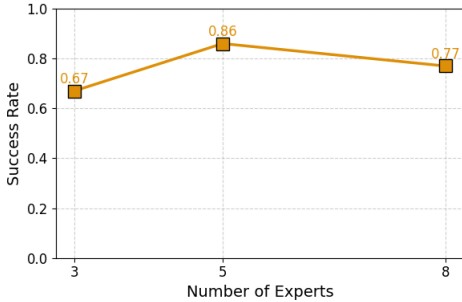

Figure 9: Performance comparison with different numbers of experts.

Table 10: Performance for different expert number.

| Method | Rearrange cup | Pour drink | Fold towel | Push objects | Stir in cup | Pick and place into basket | Throw into trash | Open drawer | Close drawer | Total |
|---|---|---|---|---|---|---|---|---|---|---|
| expert number=3 | 0.60 | 0.70 | 0.70 | 0.70 | 0.70 | 0.80 | 0.40 | 0.80 | 0.60 | 0.67 |
| expert number=5 | 0.80 | 1.00 | 1.00 | 0.80 | 0.80 | 1.00 | 0.60 | 0.90 | 0.80 | 0.86 |
| expert number=8 | 0.70 | 0.80 | 0.80 | 0.60 | 0.90 | 1.00 | 0.60 | 0.80 | 0.70 | 0.77 |

# B STATEMENT

## B.1 THE USE OF LARGE LANGUAGE MODELS (LLMS)

We use LLMs only to fix grammar errors and polish writing.

