# OpenReview forum: "Learning Diverse Skills for Behavior Models with Mixture of Experts"
_ICLR.cc/2026/Conference — Submitted to ICLR 2026_

### Official Review · Reviewer_o12B · 2025-10-24

**Soundness:** 2
**Presentation:** 3
**Contribution:** 3
**Rating:** 4
**Confidence:** 4

**Summary:**

The paper proposes Di-BM, a method integrating a Mixture of Experts (MoE) into the robotic learning framework, using an Energy-Based Model (EBM) to route tasks to suitable experts. The primary goal is to address the performance degradation and "averaging effect" of existing imitation learning models in multi-task settings. The authors propose an MoE structure that enables experts to specialize in sub-regions of the observation space, and which can be easily integrated into existing imitation learning methods as a "plug-and-play" module. The paper conducts comprehensive real-world experiments and analyses to show the effectiveness of the approach, demonstrating strong multi-task performance and superior data efficiency when fine-tuning.

**Strengths:**

- The authors performed a thorough empirical investigation, including key ablation studies on MoE formulation and the number of experts, to show how their proposed approach compares against baselines.
- Di-BM performs well across multiple real-world manipulation tasks, indicating that the proposed diverse skill learning approach can effectively address the "averaging effect" in a challenging multi-task environment.

**Weaknesses:**

- The discussion of related work on MoE in robotics appears to be missing several relevant, recent methods [1, 2, 3]. It would be difficult to determine the precise contribution and novelty of Di-BM without discussing how it compares to or differs from these existing approaches.
- The experiment settings could be clearer. For instance, the paper's central claim is about 'multi-task training', but the exact training procedure isn't explicitly defined. The authors should clarify if "multi-task training" means a single policy was trained on an aggregated dataset containing all 9 real-world manipulation tasks, as this is critical to evaluating the claims about mitigating the "averaging effect".
- The real-world experiment results are based on "10 trials per task". This sample size may be too small to robustly justify the effectiveness of the approach, as the difference between, for example, a 0.80 and 0.60 success rate (as seen in Table 1) might not be statistically significant. The authors should consider running more trials or, at a minimum, acknowledging this as a limitation.

[1] Efficient diffusion transformer policies with mixture of expert denoisers for multitask learning

[2] Mixture-of-experts network with task-oriented perturbation for visual reinforcement learning

[3] Sparse diffusion policy: A sparse, reusable, and flexible policy for robot learning

**Questions:**

Regarding the discussion in Section 4.1:
1. The paper states that the action entropy bonus $H(\pi(a|o))$ is omitted from Equation (2). However, this term was not present in Equation (2) as defined in Section 3.2. Could the authors please clarify this discrepancy in the presentation?
2. Can the author elaborate on how the diffusion model's architecture specifically replaces the function of this entropy bonus, as it is a key design choice?

---

> ### Author Response · Authors · 2025-11-21
>
> > **Response for Weakness 1**: The discussion of related work on MoE in robotics appears to be missing several relevant, recent methods [1, 2, 3]. It would be difficult to determine the precise contribution and novelty of Di-BM without discussing how it compares to or differs from these existing approaches.
>
> Thank you for your valuable comments. We include citation and discussion of [1, 2, 3] in Section~2.1 of the revised version. [2] introduces MoE and a task-oriented perturbation mechanism for visual reinforcement learning. [3] uses a separate router for each task with shared experts across tasks, while [1] conditions the MoE router on the diffusion noise level. These methods primarily rely on manually defined task categories or on routers that learn to assign probabilities to experts.
>
> Distinct from the above approaches, our method leverages EBM-based observation distributions together with a gating network to automatically allocate experts to their preferred domains based on their performance over different observations. This enables the experts to specialize and learn primitive skills without requiring explicit task-level supervision.
>
> In addition, we add comparisons with MoDE[1] and SDP[3] in Section~5.2 of the revised version, where we conduct simulation experiments on the RoboTwin 2.0 [4] benchmark. The results are presented in the table below. We observe that our method, Di-BM, significantly outperforms all established baselines in multi-task learning. We believe that these additional comparisons allow for a clearer assessment of the precise contribution and novelty of Di-BM.
>
> | Method | Adjust bottle | Beat block hammer | Click alarmclock | Dump bin bigbin | Grab roller | Handover block | Handover mic | Lift pot | Total |
> |--------|---------------|-----------------|-----------------|----------------|------------|----------------|-------------|---------|-------|
> | DP     | 0.82          | 0.06            | 0.85            | 0.55           | **0.95**   | 0.31           | 0.52        | 0.83    | 0.61  |
> | SDP [3]| **0.95**      | 0.15            | 0.73            | 0.35           | 0.79       | 0.05           | 0.42        | 0.24    | 0.46  |
> | MoDE [5]| 0.82         | 0.15            | 0.73            | 0.59           | 0.89       | 0.33           | 0.56        | 0.68    | 0.59  |
> | Di-BM  | 0.83          | **0.41**        | **0.88**        | **0.60**       | 0.92       | **0.72**       | **0.78**    | **0.91**| **0.76** |
>
> > **Response for Weakness 2**: The experiment settings could be clearer. For instance, the paper's central claim is about 'multi-task training', but the exact training procedure isn't explicitly defined. The authors should clarify if "multi-task training" means a single policy was trained on an aggregated dataset containing all 9 real-world manipulation tasks, as this is critical to evaluating the claims about mitigating the "averaging effect".
>
> Thank you for your comments. Yes, in our setting, “multi-task training’’ indicates that **a single policy is trained on an aggregated dataset** containing demonstrations from all 9 manipulation tasks. And the trained policy is subsequently evaluated on each individual task, as our goal is to enable a single model to acquire multiple skills while mitigating the “averaging effect. We have clarified this training procedure in Section~6.1 of the revised version.
>
> > **Response for Weakness 3**: The real-world experiment results are based on "10 trials per task". This sample size may be too small to robustly justify the effectiveness of the approach, as the difference between, for example, a 0.80 and 0.60 success rate (as seen in Table 1) might not be statistically significant. The authors should consider running more trials or, at a minimum, acknowledging this as a limitation.
>
> Thank you for your valuable comments. We acknowledge that 10 trials per task in the real-world experiments may be too small. Due to the practical challenges and high cost of running real-world robot experiments, we complement the results with extensive simulation evaluations on the RoboTwin2.0 [4] benchmark and we conduct 100 trials per task in simulation. As shown in the table provided in the reply to the first weakness and in Section~5.2 of the revised version, our method improves success rate by approximately 15% over the baselines. We believe that these additional experiments offer a more robust evaluation of the effectiveness of our approach.
>
>
> [1] Efficient diffusion transformer policies with mixture of expert denoisers for multitask learning
>
> [2] Mixture-of-experts network with task-oriented perturbation for visual reinforcement learning
>
> [3] Sparse diffusion policy: A sparse, reusable, and flexible policy for robot learning
>
> [4] Robotwin 2.0: A scalable data generator and benchmark with strong domain randomization for robust bimanual robotic manipulation
>
> [5] Efficient Diffusion Transformer Policies with Mixture of Expert Denoisers for Multitask Learning.

---

> ### Author Response · Authors · 2025-11-21
>
> > **Response for Question 1**: The paper states that the action entropy bonus ${H(\pi(a|o))}$ is omitted from Equation (2). However, this term was not present in Equation (2) as defined in Section 3.2. Could the authors please clarify this discrepancy in the presentation?
>
> Thank you for pointing this out. For clarity, our intention is to express that the action entropy bonus ${H(\pi(a|o))}$ is already omitted in Equation(2) due to the use of the diffusion model and for simplicity of presentation. In detail, our approach integrates multi-skill learning with a diffusion policy (DP), which naturally models the multimodal action distributions present in human demonstrations. As a result, an explicit entropy term is unnecessary. Removing this term produces a cleaner and more stable optimization objective for both the experts and the gating network. We have clarified this explicitly in Section~3.2 of the revised version.
>
> > **Response for Question 2**: Can the author elaborate on how the diffusion model's architecture specifically replaces the function of this entropy bonus, as it is a key design choice?
>
> According to [1, 2], the entropy bonus encourages the policy to learn diverse solutions. In our method, however, we employ a diffusion model architecture, which has been shown to naturally capture multimodal action distributions in human demonstrations [3]. In other words, the diffusion model inherently learns diverse solutions without requiring an explicit entropy bonus term.
>
> Furthermore, omitting this entropy term simplifies the optimization of both the expert networks and the gating network, avoiding additional terms introduced by this bonus. This is also one of the reasons why we remove it in our formulation.
>
> [1] Specializing versatile skill libraries using local mixture of experts
>
> [2] Acquiring diverse skills using curriculum reinforcement learning with mixture of experts
>
> [3] Diffusion policy: Visuomotor policy learning via action diffusion

---

> > ### Comment · Reviewer_o12B · 2025-11-26
> >
> > Thank you for the detailed clarifications and the additional comparisons and experiments. The revisions address all my concerns, and I will update my score accordingly.

---

> > > ### Author Response · Authors · 2025-11-27
> > >
> > > Thank you for your thoughtful reassessment and for being willing to raise the score—we sincerely appreciate it.

---

### Official Review · Reviewer_tBc1 · 2025-10-27

**Soundness:** 3
**Presentation:** 4
**Contribution:** 3
**Rating:** 6
**Confidence:** 4

**Summary:**

This paper proposes **Di-BM**, a novel Mixture of Experts (MoE) framework designed to improve multi-task performance in robotic imitation learning. The key problem it addresses is the "averaging effect" that plagues policies trained on diverse datasets.
It utilizes Energy-Based Models (EBMs) as the gating mechanism, which models each expert's favored observation distribution.
This allows experts to autonomously specialize in sub-regions of the observation space, which correspond to primitive skills.
The method is implemented as a plug-in for Diffusion Policy (DP) and evaluated on 9 real-world manipulation tasks. The results show significant improvements in success rates over the valinla DP baseline and MoE variants.
Furthermore, the paper demonstrates that the pre-trained Di-BM model exhibits superior data efficiency when fine-tuned on novel tasks.

**Strengths:**

1. **Novel Direction**: The core idea of applying MoE to imitation learning is novel.

2. **Strong Empirical Results**: The paper provides strong quantitative evidence for its claims. The performance leap over the baselines is substantial. The real-world experiments demonstrate the potential of practical applications.

3. **Analysis of Learned Experts**: The visualization of expert activation $\pi(e|o)$ in Fig. 3 supports the hypothesis that the experts are specializing in distinct phases or primitive skills that are shared across different tasks.

4. **Data Efficiency for Fine-Tuning**: The fine-tuning experiment in Sec. 5.6 shows that the pre-trained Di-BM model adapts to new tasks more efficiently than baselines. This suggests the learned primitive skills are reusable and indicates potential for future scalability.

5. **Clarity**: The paper is well-written and easy to follow. The methodology and motivation are clearly explained, supported by effective figures and visualizations.

**Weaknesses:**

1. **Limited Methodological Novelty**: Although applying MoE to imitation learning is novel, the core methodology follows the prior work of MoE in RL [1].

2. **Sensitivity to Hyperparameter**: The authors mention the model's sensitivity to the KL regularization coefficient $\beta$ (Sec. 5.5, Fig. 4). A $\beta$ that is too small causes all experts to "slack off" and avoid difficult parts of the observation space. This sensitivity could present challenges when scaling Di-BM to more complex tasks or datasets.

3. **Missing Analysis of Computational Overhead**: The paper claims "minimal computational overhead" without explicit analysis. At inference, the model must first compute the expert probabilities via the gating network $g_{\phi}$ before executing the selected expert $f_{\theta}$. Especially in robotics applications, where real-time inference and reaction speed are crucial. An analysis of inference-time overhead is important.

**Questions:**

1. Figure 3 shows that the router utilizes different experts as a task progresses. How can we be sure the router is learning to select the *best* expert for each stage? The paper shows performance for individual experts on *full* tasks (Table 3), but it would be more convincing to see an analysis of individual expert performance on manually-partitioned task stages to verify the router's choices.

2. How does the routing strategy evolve during training? One would expect experts to have similar weights initially and then specialize. Is this specialization and the resulting routing strategy consistent across different training runs with different random seeds? In other words, do the experts learn the same set of "primitives" each time?

3. In Sec. A.6, the authors state that when using 8 experts, some are underutilized or "pruned". If these experts are simply pruned, why does the model's overall performance drop substantially (Table 8)  instead of matching the 5-expert model? Furthermore, Figure 3 also shows that some experts are underutilized. Given this, would it be possible to develop a method that dynamically adjusts or prunes the number of experts during training?

4. The paper's main novelty is adapting the Di-SkilL [1] framework from RL to IL. Could you elaborate on the specific challenges encountered and design choices in this adaptation?

I am willing to raise my score if my concerns are addressed.

[1] Celik, Onur, Aleksandar Taranovic, and Gerhard Neumann. "Acquiring diverse skills using curriculum reinforcement learning with mixture of experts." ICML, 2024.

---

> ### Author Response · Authors · 2025-11-21
>
> > **Response for Weakness 1**: Limited Methodological Novelty: Although applying MoE to imitation learning is novel, the core methodology follows the prior work of MoE in RL.
>
> Thank you for your comments. We agree that our work is inspired by the multi-skill learning paradigm in[1], but we respectfully emphasize several key differences that introduce substantial methodological novelty and empirical advancement:
>
> **Scalability to High-Dimensional, Real-World Tasks.**
> Prior works evaluate minimalist architectures on simple simulated tasks (e.g., 5-Link Reacher, Table Tennis). In contrast, we integrate the MoE paradigm into the Diffusion Policy framework and demonstrate that it scales effectively to complex, high-dimensional real-world manipulation tasks. We view this successful integration and scaling as a non-trivial and impactful contribution to the robot learning community.
>
> **Architectural and Training Differences.**
>
>    * We introduce extensive parameter sharing to improve efficiency and generalization: all experts share the observation encoder and parts of the action model, and the gating network also shares the observation encoder. This design differs fundamentally from the independent expert architectures in [1]. Parameter sharing allows experts and the gating network to benefit from common representations, improves computation efficiency, and facilitates transfer of shared knowledge across skills [3, 4]—an important consideration in real-world manipulation tasks with limited data.
>
>    * [1] incorporates the entropy bonus in the optimization objective to encourage diverse solutions. In contrast, our approach integrates multi-skill learning with diffusion policy (DP) [5], which inherently models multimodal action distributions present in human demonstrations, rendering an explicit entropy term unnecessary. Eliminating this term produces a cleaner and more stable optimization objective. This adjustment better aligns the framework with DP-based imitation learning and simplifies training for both the experts and the gating network (see revised Section 3.2).
>
>    * While [1] updates experts sequentially and separates expert and gating network updating, we find that such a scheme becomes highly unstable in high-capacity and shared parameter settings. To address this, we adopt a simultaneous update scheme that jointly optimizes all experts and the gating network. This modification is critical for achieving stable and reliable training in large imitation-learning models. We have included a more detailed explanation of this necessary modification to the training procedure in Section 4.3.1 of the revised version.
>
> > **Response for Weakness 2**: Sensitivity to Hyperparameter: The authors mention the model's sensitivity to the KL regularization coefficient β (Sec. 5.5, Fig. 4). A $\beta$ that is too small causes all experts to "slack off" and avoid difficult parts of the observation space. This sensitivity could present challenges when scaling Di-BM to more complex tasks or datasets.
>
> Thank you for your comments. To further investigate the effect of β on model performance, we conduct an ablation study on the RoboTwin 2.0 [2] benchmark, and the results are shown in the table below. We observe that a large β over-constrains the experts and prevents them from specializing in their respective domains, leading to behavior similar to a standard diffusion policy. In contrast, a very small $\beta$ allows experts to ``slack off'' on the harder parts of the task, resulting in degraded performance.
>
> These findings provide practical guidance for applying Di-BM to more complex tasks or datasets: it is generally safer to start with a relatively large β, which avoids catastrophic performance collapse, and then gradually decrease β if needed. We agree that reducing sensitivity to this hyperparameter is desirable, and we plan to explore mechanisms for dynamically adapting β in future work.
>
> | β     | Avg. Success |
> |-------|--------------|
> | 0.001 | 0.23         |
> | 0.01  | **0.76**     |
> | 0.1   | 0.59         |
> | 1     | 0.60         |
> | 10    | 0.63         |
>
> [1] Acquiring diverse skills using curriculum reinforcement learning with mixture of experts. ICML, 2024.
>
> [2] Robotwin 2.0: A scalable data generator and benchmark with strong domain randomization for robust bimanual robotic manipulation. arXiv preprint, 2025.
>
> [3] Adashare: Learning what to share for efficient deep multi-task learning. Advances in Neural Information Processing Systems, 2020.
>
> [4] Task adaptive parameter sharing for multi-task learning. Proceedings of the IEEE/CVF Conference on Computer Vision and Pattern Recognition, 2022.
>
> [5] Diffusion policy: Visuomotor policy learning via action diffusion. The International Journal of Robotics Research, 2025.

---

> ### Author Response · Authors · 2025-11-21
>
> > **Response for Weakness 3**: Missing Analysis of Computational Overhead: The paper claims "minimal computational overhead" without explicit analysis. At inference, the model must first compute the expert probabilities via the gating network  before executing the selected expert. Especially in robotics applications, where real-time inference and reaction speed are crucial. An analysis of inference-time overhead is important.
>
> Thank you for your comments. Our claim of “minimal computational overhead’’ refers to the fact that although introducing MoE layers increases the total parameter count, the actual computation per forward pass remains low because only a sparse subset of experts is activated during inference. In our setting, the router selects only the top-1 expert, so the computation does not scale with the total number of experts. This property is well recognized in classical MoE works[1, 2].
> In addition, the introduced gating network is a lightweight 3-layer MLP that shares the same observation encoder as the action model. As a result, the extra computation introduced by the gating network is almost negligible.
>
> To quantify this, we measure the runtime of a standard Diffusion Policy and our Di-BM on an RTX 4090 GPU (with 16 denoising steps) in real-world deployment. Their inference times are 42ms and 45ms, respectively. Overall, Di-BM does not introduce significant computational cost and supports real-time execution. We include the runtime analysis in Section~6.2 of the revised version.
>
> > **Response for Question 1**: Figure 3 shows that the router utilizes different experts as a task progresses. How can we be sure the router is learning to select the best expert for each stage? The paper shows performance for individual experts on full tasks (Table 3), but it would be more convincing to see an analysis of individual expert performance on manually-partitioned task stages to verify the router's choices.
>
> Although evaluating individual expert performance on manually-partitioned task stages is challenging—since task execution is sequential and failures in the initial stages often lead to overall task failure—we can provide some qualitative observations from our tests. From Figure 3, we observe that Expert 0 has high selection probability at the beginning and end of the *fold towel* task, but lower probability in the middle stage. In testing, we observe that using only Expert 0 often causes the towel to be initially grasped and then released, leading to task failure. Similarly, from Figure 3, we observe that Expert 3 has low selection probability in the *push objects* task, especially at the beginning. In testing, using only Expert 3 often results in the gripper failing to align with the object to be pushed, causing task failure.
>
> > **Response for Question 2**: How does the routing strategy evolve during training? One would expect experts to have similar weights initially and then specialize. Is this specialization and the resulting routing strategy consistent across different training runs with different random seeds? In other words, do the experts learn the same set of "primitives" each time?
>
> Thank you for the insightful question. We observe that the routing strategy does not converge to an identical configuration across different random seeds. While experts consistently specialize over the course of training—starting from nearly uniform routing weights and gradually developing distinct roles—the exact allocation of primitive skills to experts varies across runs. Despite these differences in assignment, different expert–primitive mappings lead to comparable success rates, indicating the decomposition is flexible rather than brittle.
>
>
> [1] Switch transformers: Scaling to trillion parameter models with simple and efficient sparsity. Journal of Machine Learning Research, 2022.
>
> [2] Deepseekmoe: Towards ultimate expert specialization in mixture-of-experts language models. arXiv preprint, 2024.

---

> ### Author Response · Authors · 2025-11-21
>
> > **Response for Question 3**: In Sec. A.6, the authors state that when using 8 experts, some are underutilized or "pruned". If these experts are simply pruned, why does the model's overall performance drop substantially (Table 8) instead of matching the 5-expert model? Furthermore, Figure 3 also shows that some experts are underutilized. Given this, would it be possible to develop a method that dynamically adjusts or prunes the number of experts during training?
>
> Thank you for the comment. The observed performance drop in the 8-expert setting (Table 8) is likely due to suboptimal tuning of the KL regularization coefficient β. When additional experts are added without appropriately adjusting β, some experts may remain underutilized while others fail to specialize effectively, introducing mild interference and slightly degrading overall performance. We agree that dynamically adjusting the number of experts or the β parameter during training is a promising direction. Such a mechanism could allow the model to prune or reallocate underutilized experts while maintaining specialization, potentially improving efficiency and robustness. We will explore this direction in future work.
>
> > **Response for Question 4**: The paper's main novelty is adapting the Di-SkilL framework from RL to IL. Could you elaborate on the specific challenges encountered and design choices in this adaptation?}
>
> Please see the reply for the first weakness.

---

### Official Review · Reviewer_myRN · 2025-10-27

**Soundness:** 2
**Presentation:** 3
**Contribution:** 2
**Rating:** 4
**Confidence:** 4

**Summary:**

This paper introduces Di-BM, a Mixture of Experts (MoE) framework designed to improve multi-task imitation learning for robotic manipulation. Traditional behavior models trained on diverse demonstrations suffer from task interference and “averaging effects.” Di-BM addresses this by associating each expert with a distinct observation distribution, modeled via an energy-based model. The gating network automatically allocates data to the most suitable expert, allowing each to specialize in a subset of primitive skills. They represent each expert using a diffusion model and evaluate their model on real-world robotic manipulation tasks.

**Strengths:**

- Gating visualisations nicely show that the model utilises different experts
- The methods shows strong empirical results, showing improvement on several real-world robotic tasks, verified through ablations and visualizations
- The method can be incorporated seamlessly into existing imitation learning architectures

**Weaknesses:**

- The paper does not mention related work that uses very similar methodology and goals, namely [1] and [2].

- In [1] they show that the optimal gating can be computed in closed form, making it unnecessary to learn a model in every iteration but it is sufficient to only learn a gating at the end of training. What benefit do the authors see when learning the gating?

- Additionally, in [1] the authors establish convergence guarantees from an expectation-maximisation perspective. Do the authors think similar results could be applied here? As an additional comment, in e.g. [3] they show that the diffusion noise matching loss is a lower bound on the marginal likelihood. In that sense, the expert objective could be seen as a lower bound on a weighted maximum likelihood objective.

- The authors only consider real-world experiments. It would be good to also include comparisons in established simulation-based benchmarks. On that note, there exists a benchmark that is designed for diverse behaviour learning, see [4]. It would be interesting to see if the proposed method improves behaviour diversity over existing methods.

- It would be good if the authors provide an ablation study showing how sensitive the performance is with respect to the KL regularisation parameter. From Figure 4, it seems like even minor changes can have a huge impact on the learned model. How difficult is it to choose the parameter? Did the authors tune the parameter per task or did they find a setting that worked for all?

[1] Information maximizing curriculum: A curriculum-based approach for learning versatile skills
[2] Curriculum-based imitation of versatile skills
[3] Towards Diverse Behaviors: A Benchmark for Imitation Learning with Human Demonstrations
[4] Understanding Diffusion Models: A Unified Perspective

**Questions:**

See weaknesses.

**Details Of Ethics Concerns:**

None.

---

> ### Author Response · Authors · 2025-11-21
>
> >**Response for Weakness 1**: The paper does not mention related work that uses very similar methodology and goals, namely [1] and [2].
>
> We thank the reviewer for pointing out these closely related works. We have now added citation and a dedicated discussion in Section 2.1 of the revised version, where we clarify the methodological differences between our approach and [1, 2].
>
> >**Response for Weakness 2**: In [1] they show that the optimal gating can be computed in closed form, making it unnecessary to learn a model in every iteration but it is sufficient to only learn a gating at the end of training. What benefit do the authors see when learning the gating?
>
> [1] computes each expert’s curriculum ${p(o,a|z)}$ in closed form and uses it to weight the loss of every expert on every data point within each batch. This weighting-by-curriculum mechanism is conceptually related to our sampling-based expert selection. However, because an epoch contains many iterations, relying solely on the weighting computed from the current batch may discard information accumulated from previous batches. We believe this per-batch weighting strategy may exhibit instability when applied to more complex tasks. Notably, the tasks considered in [1] are relatively simple, with the Franka Kitchen environment having an observation dimension of only 30. In future work, we plan to examine how this strategy performs in more challenging settings.
>
> In contrast, our method jointly updates the experts while training a gating network that conditions on encoder representations. This allows the gating network to learn consistent expert specializations from semantic features shared across batches, rather than merely computing and storing batch-specific weights. Consequently, our approach provides a stable and semantically informed mechanism for expert allocation in complex visuomotor tasks. We argue that this design offers an important advantage: the gating network can learn the structure of the observation space and discover the implicit domains of expertise for each expert, rather than relying solely on per-batch statistics. Similar insights have been discussed in prior mixture-of-experts literature—e.g., adaptive expert specialization through learned gating has been shown to improve task partitioning and generalization in high-dimensional settings [4, 5].
>
> >**Response for Weakness 3**: Additionally, in [1] the authors establish convergence guarantees from an expectation-maximisation perspective. Do the authors think similar results could be applied here? As an additional comment, in e.g. [3] they show that the diffusion noise matching loss is a lower bound on the marginal likelihood. In that sense, the expert objective could be seen as a lower bound on a weighted maximum likelihood objective.
>
> Thank you for the insightful comments. To the best of our understanding, the convergence guarantees in [1] rely on performing updates with respect to the entire dataset at each iteration (i.e., each ``batch'' contains all data). As noted by the authors themselves in Appendix A.1 of [1], these guarantees do not strictly hold under stochastic gradient ascent. Whether such guarantees can extend to large-scale, high-dimensional visuomotor tasks with deeper neural networks remains an open question and would likely require additional empirical validation.
>
> However, both the expectation-maximisation perspective in [1] and "the diffusion noise matching loss can be viewed as a lower bound on a weighted maximum-likelihood objective for the experts" from reviewer provide valuable conceptual insights for us. We believe these perspectives are promising directions, and in future work we plan to explore how they may be combined with our learnable gating-network framework and applied to more complex real-world robotic manipulation tasks.
>
> [1] Information maximizing curriculum: A curriculum-based approach for learning versatile skills
>
> [2] Curriculum-based imitation of versatile skills
>
> [3] Understanding Diffusion Models: A Unified Perspective
>
> [4] Outrageously Large Neural Networks: The Sparsely-Gated Mixture-of-Experts Layer. ICLR, 2017.
>
> [5] Switch Transformers: Scaling to Trillion Parameter Models with Simple and Efficient Sparsity.
> JMLR, 2022.

---

> ### Author Response · Authors · 2025-11-21
>
> >**Response for Weakness 4**: The authors only consider real-world experiments. It would be good to also include comparisons in established simulation-based benchmarks. On that note, there exists a benchmark that is designed for diverse behaviour learning, see [4]. It would be interesting to see if the proposed method improves behaviour diversity over existing methods.
>
> Thank you for this valuable suggestion. According to your suggestion, we have conducted additional multi-task simulation experiments using the established RoboTwin 2.0 [1] benchmark. The results in following table show that our method consistently outperforms existing baselines on these diverse and challenging tasks. The new experimental results and detailed analysis have been added to Section 5.2 of the revised version. We believe that these simulation-based evaluations further strengthen the empirical validity of our approach.
>
> We also clarify that the notion of "diverse skills" in our work refers to learning different primitive manipulation skills (e.g., grasping, pushing, pulling, stirring), rather than producing diverse solutions under the same state. Of course, since our framework builds on Diffusion Policy, it naturally inherits the inherent behavioral diversity of DDPMs.
>
> | Method | Adjust bottle | Beat block hammer | Click alarmclock | Dump bin bigbin | Grab roller | Handover block | Handover mic | Lift pot | Avg. |
> |--------|---------------|-----------------|----------------|----------------|------------|----------------|--------------|----------|------|
> | DP     | 0.82          | 0.06            | 0.85           | 0.55           | **0.95**   | 0.31           | 0.52         | 0.83     | 0.61 |
> | SDP[2]| **0.95**      | 0.15            | 0.73           | 0.35           | 0.79       | 0.05           | 0.42         | 0.24     | 0.46 |
> | MoDE[3]| 0.82         | 0.15            | 0.73           | 0.59           | 0.89       | 0.33           | 0.56         | 0.68     | 0.59 |
> | Di-BM  | 0.83          | **0.41**        | **0.88**       | **0.60**       | 0.92       | **0.72**       | **0.78**     | **0.91** | **0.76** |
>
> >**Response for Weakness 5**: It would be good if the authors provide an ablation study showing how sensitive the performance is with respect to the KL regularisation parameter. From Figure 4, it seems like even minor changes can have a huge impact on the learned model. How difficult is it to choose the parameter? Did the authors tune the parameter per task or did they find a setting that worked for all?
>
> Thank you for your insightful comments. We conduct an ablation study on the coefficient of the KL regularization term using the RoboTwin 2.0 [1] benchmark, and the results are shown in the table below. We observe that a large $\beta$ over-constrains the experts and prevents them from specializing in their respective domains, leading to behavior similar to a standard diffusion policy. In contrast, a very small $\beta$ allows experts to "slack off" on the harder parts of the task, resulting in degraded performance. This observation further supports our hypothesis that, under an appropriate choice of $\beta$, each expert is able to specialize in specific skills. The ablation study on the KL regularization coefficient has been added to Section 5.2 of the revised version.
>
> In practice, we evaluate a series of $\beta$ values spaced logarithmically (i.e., varying by an order of magnitude). Importantly, in both simulation and real-world experiments, we train **a single model on the dataset across all tasks and test it on all tasks**, rather than tuning $\beta$ for each individual task. In simulation, $\beta = 0.01$ performs best across tasks, whereas in the real world $\beta = 0.003$ is most effective. We believe this difference mainly stems from the distinct observation encoders and dataset scales used in simulation versus real-world settings. We agree that reducing manual tuning is important, and in future work we plan to explore mechanisms for dynamically adapting $\beta$ during training.
>
> | $\beta$     | Avg. Success |
> |-------|--------------|
> | 0.001 | 0.23         |
> | 0.01  | **0.76**     |
> | 0.1   | 0.59         |
> | 1     | 0.60         |
> | 10    | 0.63         |
>
> [1] Robotwin 2.0: A scalable data generator and benchmark with strong domain randomization for robust bimanual robotic manipulation. arXiv preprint 2025.
>
> [2] Sparse Diffusion Policy: A Sparse, Reusable, and Flexible Policy for Robot Learning. Conference on Robot Learning, 2025.
>
> [3] Efficient Diffusion Transformer Policies with Mixture of Expert Denoisers for Multitask Learning. The Thirteenth International Conference on Learning Representations, 2025.

---

> > ### Comment · Reviewer_myRN · 2025-11-27
> > **Reply**
> >
> > I thank the authors for their reply. I increased my score accordingly.

---

> > > ### Author Response · Authors · 2025-11-28
> > >
> > > Thank you for updating the score — we sincerely appreciate it.

---

### Official Review · Reviewer_jt5F · 2025-10-27

**Soundness:** 3
**Presentation:** 3
**Contribution:** 2
**Rating:** 4
**Confidence:** 3

**Summary:**

This paper introduces Diverse Skills for behavior models (Di-BM), an imitation learning method that can learn policies for multi-task settings. More concretely, Di-BM employs Mixture of Experts policies that can specialize on a subset of the available training data by employing an energy-based gating network. The gating network represents an observation distribution per expert, thereby allowing each expert to specialize in observation-action regions it favors. The paper shows, on real-robot experimental tasks, that the proposed method works nicely and is able to learn multi-task policies.

**Strengths:**

- The paper is well-written. The reader can easily follow the story of the paper and understand the motivation behind the proposed methods
- The benefits of choosing an MoE policy representation are well-grounded by intuitive figures (e.g., Fig.3) on the training data
- The method is validated on real-robot experiments, emphasizing its strengths

**Weaknesses:**

- The work lacks important related works that have employed similar ideas on learning parameterized distributions over the input (observation) space [1,2]. How does the proposed method algorithmically differ from these methods?
- Better description of the data set; i.e., which tasks are included? Are different robot types used? .... It's hard to infer the "difficulty" of a task. Commenting on the task difficulty could help the reader.
- Although I appreciate the real-robot experiments, I believe the paper would be strengthened if Di-BM could be benchmarked on common benchmark suites, for example, such as LIBERO [3] or Robocasa [4].

[1] M. X. Li, et al. Curriculum-based imitation of versatile skills. ICRA 2023.
[2] D. Blessing, et al. Information Maximizing Curriculum: A Curriculum-Based Approach for Imitating Diverse Skills. NeurIPS 2023.
[3] O. Mees, et al. Calvin: A benchmark for language-conditioned policy learning for long-horizon robot manipulation tasks. RA-L 2022.
[4] B. Liu, et al. Libero: Benchmarking knowledge transfer for lifelong robot learning. NeurIPS 2023.

**Questions:**

- In Section 4.2 it is said that the observations $o \sim p(o)$ are sampled to cover a sufficiently large batch of observations. Does this mean the observations are sampled from the offline data set, or do the observations come from the environment by sampling from it?
If the observations come from the data set, how does the method behave if we can not sample a representative batch of observations for approximating the normalization constant? Or in general, was this a problem observed during the experiments?

- It seems that single experts are even better than the baselines from Table 2. For example, all experts from Table 3 are better on the Rearrange cup task than the Task-wise MoE model from Table 2. Is there an intuitive explanation behind it? Intuitively, I would have expected that the Task-wise MoE performs better consistently compared to the single experts.

- Does the MoE integration also work for goal-conditioned, score-based diffusion policies[5]?


[5] M. Reuss, et al. Goal-conditioned Imitation Learning using Score-based Diffusion Policies. RSS 2023.

---

> ### Author Response · Authors · 2025-11-21
>
> >**Response for Weakness 1**: The work lacks important related works that have employed similar ideas on learning parameterized distributions over the input (observation) space [1,2]. How does the proposed method algorithmically differ from these methods?
>
> We appreciate the reviewer bringing these important related works to our attention. We have added citation and a dedicated discussion comparing our approach with [1, 2] in Section 2.1 of the revised version.
>
> Algorithmically, our core difference lies in the mechanism used to determine the contribution or selection probability of each expert (or component). The methods in [1, 2] determine the contribution of each expert component by assigning a weight to each sample for each expert during training, which can be calculated using a closed-form solution based on the data in the current batch.
>
> In contrast, our approach replaces this closed-form assignment with a learnable gating network that is jointly optimized with all experts. We argue that this design offers an important advantage: the gating network can learn the structure of the observation space and discover the implicit domains of expertise for each expert, rather than relying solely on per-batch statistics. Similar insights have been discussed in prior mixture-of-experts literature—e.g., adaptive expert specialization through learned gating has been shown to improve task partitioning and generalization in high-dimensional settings [3, 4]. Building on these findings, our gating network allows the model to automatically align each expert with relevant regions of the observation space in a data-driven manner.
>
> >**Response for Weakness 2**: Better description of the data set; i.e., which tasks are included? Are different robot types used? .... It's hard to infer the "difficulty" of a task. Commenting on the task difficulty could help the reader.
>
> Thank you for your comments. We have thoroughly revised the manuscript to include this necessary information.
> * **Included Tasks.** Our training dataset includes 9 distinct manipulation tasks: *rearrange cup, pour drink, fold towel, push objects, stir in cup, pick and place into basket, throw into trash, open drawer, close drawer*. The model is also tested on all these 9 tasks. Details have been added to Section 6.1 of the revised version.
>
> * **Robot Types (data Collection and testing)**
>   * **Data Collection:** The training data was collected using the UMI system (a hand-held gripper system designed for efficient data acquisition) by human, not the final robotic arm used in test.
>   * **Real-World Testing:** The actual real-world experiments and policy deployment were performed on the Nova5 robot arm. We have added explanation for this in Section 6.2 of the revised version.
>
> * **Task Difficulty.** To help the reader infer difficulty, we categorize the tasks into “easy” and “hard” categories based on (i) required manipulation precision and (ii) multi-step complexity. In addition, inspired by human motor-skill developmental curves discussed in [5], we provide a difficulty assessment following established benchmarks in robotic manipulation literature [5]. Tasks such as *throw into trash, open drawer, close drawer, fold towel, push objects, pick and place into basket* are categorized as "easy", while tasks like *rearrange cup, stir in cup, and pour drink* are categorized as "hard". We have added relevant explanations in Tables 3 and 4 of the revised version. For specific details on the tasks, please refer to Appendix A.3.
>
> [1] M. X. Li, et al. Curriculum-based imitation of versatile skills. ICRA 2023.
>
> [2] D. Blessing, et al. Information Maximizing Curriculum: A Curriculum-Based Approach for Imitating Diverse Skills. NeurIPS 2023.
>
> [3] Outrageously Large Neural Networks: The Sparsely-Gated Mixture-of-Experts Layer. ICLR, 2017.
>
> [4] Switch Transformers: Scaling to Trillion Parameter Models with Simple and Efficient Sparsity.
> JMLR, 2022.
>
> [5] Towards Human-Level Bimanual Dexterous Manipulation with Reinforcement Learning. NeurIPS, 2022.

---

> ### Author Response · Authors · 2025-11-21
>
> >**Response for Weakness 3**: Although I appreciate the real-robot experiments, I believe the paper would be strengthened if Di-BM could be benchmarked on common benchmark suites, for example, such as LIBERO or Robocasa.
>
> Thank you for this valuable suggestion. According to your suggestion, we have conducted comprehensive multi-task simulation experiments using the established RoboTwin 2.0 [1] benchmark. The results in following table clearly show that our proposed method outperforms all existing baselines in multi-task learning tasks. These experimental results and a detailed analysis are now included in Section 5.2 of the revised version. We believe that the additional simulation experiments further strengthen the paper.
>
> | Method | Adjust bottle | Beat block hammer | Click alarmclock | Dump bin bigbin | Grab roller | Handover block | Handover mic | Lift pot | Avg. |
> |--------|---------------|-----------------|----------------|----------------|------------|----------------|--------------|----------|------|
> | DP     | 0.82          | 0.06            | 0.85           | 0.55           | **0.95**   | 0.31           | 0.52         | 0.83     | 0.61 |
> | SDP[2]| **0.95**      | 0.15            | 0.73           | 0.35           | 0.79       | 0.05           | 0.42         | 0.24     | 0.46 |
> | MoDE[3]| 0.82         | 0.15            | 0.73           | 0.59           | 0.89       | 0.33           | 0.56         | 0.68     | 0.59 |
> | Di-BM  | 0.83          | **0.41**        | **0.88**       | **0.60**       | 0.92       | **0.72**       | **0.78**     | **0.91** | **0.76** |
>
> >**Response for Question 1**: In Section 4.2 it is said that the observations ${o \sim p(o)}$ are sampled to cover a sufficiently large batch of observations. Does this mean the observations are sampled from the offline data set, or do the observations come from the environment by sampling from it? If the observations come from the data set, how does the method behave if we can not sample a representative batch of observations for approximating the normalization constant? Or in general, was this a problem observed during the experiments?}}
>
> The observations are sampled from the offline dataset. Specifically, when approximating the normalization constant, we randomly sample several batches from the offline dataset and compute the average over these samples. In practice, we find that the estimated normalization constant converges very quickly—typically within only 3–4 batches (with a batch size of 256). Thus, sampling only a few hundred observations from the offline dataset is sufficient to obtain a stable estimate. Therefore, this issue does not become a problem in our experiments.
>
>
> >**Response for Question 2**: It seems that single experts are even better than the baselines from Table 2. For example, all experts from Table 3 are better on the Rearrange cup task than the Task-wise MoE model from Table 2. Is there an intuitive explanation behind it? Intuitively, I would have expected that the Task-wise MoE performs better consistently compared to the single experts.
>
> Thank you for your observation. For the task-wise MoE model, we manually assign each expert to one or more specific task. There are a total of 9 tasks, but the number of experts in the task-wise MoE model is set to 6 (which is even larger than the number of experts-5-used by our Di-BM). Consequently, one expert is shared by both the *Rearrange cup* and *Fold towel* tasks (as detailed in Appendix A.4). This overlap introduces unintended interference between tasks, which causes a degradation in performance.
>
> In contrast, our approach expects each expert to learn different stages of tasks, or equivalently, different primitive skills. Although each expert learns from multiple tasks, the acquired skills tend to be complementary rather than conflicting. This provides an intuitive explanation for why the single experts outperform the task-wise MoE model in this case.
>
> >**Response for Question 3**: Does the MoE integration also work for goal-conditioned, score-based diffusion policies[4]?
>
> Yes, our MoE method can be applied to any imitation learning framework. We plan to explore integrating Di-BM with goal-conditioned, score-based diffusion policies [4] as well as other diffusion-based behavior cloning methods in future work. We have added a citation to [4] in Section 2.2 of the revised version.
>
> [1] Robotwin 2.0: A scalable data generator and benchmark with strong domain randomization for robust bimanual robotic manipulation. arXiv preprint 2025.
>
> [2] Sparse Diffusion Policy: A Sparse, Reusable, and Flexible Policy for Robot Learning. Conference on Robot Learning, 2025.
>
> [3] Efficient Diffusion Transformer Policies with Mixture of Expert Denoisers for Multitask Learning. The Thirteenth International Conference on Learning Representations, 2025.
>
> [4] M. Reuss, et al. Goal-conditioned Imitation Learning using Score-based Diffusion Policies. RSS 2023.

---

### Official Review · Reviewer_43vA · 2025-11-01

**Soundness:** 2
**Presentation:** 3
**Contribution:** 2
**Rating:** 4
**Confidence:** 3

**Summary:**

The paper proposes an MoE architecture and training method for robotic multi-task imitation learning. To mitigate multi-task gradient interference, a gating network is used to map the observation to the distribution of experts, and each expert policy is trained to handle a specific subset of observations. The training framework can adaptively learn the gating network using energy-based models, rather than predefine task assignment. Experiments on some real-world manipulation tasks demonstrate its effectiveness compared with previous MoE strategies.

**Strengths:**

- The MoE framework introduced in this paper can learn task assignment autonomously, enhancing the potential to learn from large-scale, unstructured multi-task datasets.
- Experimental results on real-robot tasks demonstrate its effectiveness. Qualitative visualizations show that the models do learn some meaningful task assignments.
- The paper writing is well organized and easy to follow.

**Weaknesses:**

- Limited technical contributions. After reading the method sections, it seems that most of techniques are adopted from the two prior works (Celik et al., 2022; 2024). The main difference is changing their reinforcement learning setting to the imitation learning setting.
- Lack of reproducible simulation experiments. There are a lot of multi-task imitation learning benchmarks in simulation that are widely used in prior robotic imitation learning research, like Meta-World, Libero, RoboCasa, and RoboTwin. However, the paper only uses real-world experiments, making the results less reproducible.
- The effectiveness on large-scale datasets, like Open-X-Embodiments, has not been studied.

**Questions:**

- What are the main technical differences between the proposed method and Celik et al., 2022; 2024?

---

> ### Author Response · Authors · 2025-11-21
>
> >**Response for Weakness 1**: Limited technical contributions. After reading the method sections, it seems that most of techniques are adopted from the two prior works (Celik et al., 2022; 2024). The main difference is changing their reinforcement learning setting to the imitation learning setting.
>
> Thank you for your comments. We agree that our work is fundamentally inspired by the multi-skill learning paradigm presented in Celik et al. (2022; 2024), yet we respectfully highlight several key differences that constitute significant technical novelty and empirical advancement:
>
> - **Complexity and Scalability of Application.** The prior works primarily focused on minimalist network architectures and were only validated on simple, simulated "toy" tasks (e.g., 5-Link Reacher, Table Tennis). In contrast, our work successfully integrates the multi-skill learning approach into the Diffusion Policy framework and, crucially, demonstrates its effectiveness on complex, real-world robotic manipulation tasks. We believe that such integration and successful scaling to a high-dimensional, real-world setting is non-trivial and represents a major empirical contribution for the robot learning community.
>
> - **Network Architecture and Training Methodology**
>
>   * We introduce extensive parameter sharing to enhance efficiency and generalization. Specifically, different expert networks share the observation encoder and certain layers of the action model. Additionally, the experts and the gating network share the encoder. This sharing structure is distinct from the independent architectures typically used in Celik et al. (2022; 2024). Parameter sharing allows experts and gating network to benefit from common representations, improves computation efficiency, and facilitates transfer of shared knowledge across skills [2, 3]—an important consideration in real-world manipulation tasks with limited data.
>
>
>   * Celik et al. (2022; 2024) included the entropy bonus in the optimization objective to promote policy diversity within RL. In our setting, we combine multi-skill learning with diffusion policy (DP)[1], which naturally captures multimodal action distributions from human demonstrations. This makes the entropy term unnecessary, and removing it produces a cleaner and more stable optimization objective. This adjustment better aligns the framework with DP-based imitation learning and simplifies training for both the experts and the gating network (see revised Section 3.2).
>
>
>   * Celik et al. (2022; 2024) updated the network parameters sequentially, updating the experts sequentially and separating the updates for the expert and gating networks. However, in high-capacity and shared parameter setting, we found that this sequential update scheme leads to significant training instability. To overcome this, we introduce a simultaneous update scheme where all expert networks and the gating network are optimized jointly. This modification is critical for achieving stable and reliable training in large imitation-learning models. We have included a more detailed explanation of this necessary modification to the training procedure in Section 4.3.1 of the revised version.
>
> [1] Diffusion policy: Visuomotor policy learning via action diffusion. The International Journal of Robotics Research, 2025.
>
> [2] Adashare: Learning what to share for efficient deep multi-task learning. Advances in Neural Information Processing Systems, 2020.
>
> [3] Task adaptive parameter sharing for multi-task learning. Proceedings of the IEEE/CVF Conference on Computer Vision and Pattern Recognition. 2022.

---

> ### Author Response · Authors · 2025-11-21
>
> >**Response for Weakness 2**: Lack of reproducible simulation experiments. There are a lot of multi-task imitation learning benchmarks in simulation that are widely used in prior robotic imitation learning research, like Meta-World, Libero, RoboCasa, and RoboTwin. However, the paper only uses real-world experiments, making the results less reproducible.
>
> Thank you for this valuable and constructive suggestion. We wholeheartedly agree that incorporating experiments in widely-used simulation environments is crucial for enhancing the reproducibility of research findings.
> To address this concern, we have incorporated comprehensive multi-task simulation experiments on the established RoboTwin 2.0 [1] benchmark, as suggested.
> The experimental results in following table strongly demonstrate that our proposed method significantly outperforms all existing baselines.
> These newly added simulation results further validate the effectiveness and superiority of our method in complex multi-skill imitation learning scenarios, and improve the reproducibility of our results.
> The relevant experimental results and detailed analysis have been included in Section 5.2 of the revised version.
> | Method | Adjust bottle | Beat block hammer | Click alarmclock | Dump bin bigbin | Grab roller | Handover block | Handover mic | Lift pot | Avg. |
> |--------|---------------|-----------------|----------------|----------------|------------|----------------|--------------|----------|------|
> | DP     | 0.82          | 0.06            | 0.85           | 0.55           | **0.95**   | 0.31           | 0.52         | 0.83     | 0.61 |
> | SDP [2]| **0.95**      | 0.15            | 0.73           | 0.35           | 0.79       | 0.05           | 0.42         | 0.24     | 0.46 |
> | MoDE [3]| 0.82         | 0.15            | 0.73           | 0.59           | 0.89       | 0.33           | 0.56         | 0.68     | 0.59 |
> | Di-BM  | 0.83          | **0.41**        | **0.88**       | **0.60**       | 0.92       | **0.72**       | **0.78**     | **0.91** | **0.76** |
>
> >**Response for Weakness 3**: The effectiveness on large-scale datasets, like Open-X-Embodiments, has not been studied.
>
> Thank you for your comments. Prior works have shown that large-scale datasets, such as Open-X-Embodiments, substantially enhance robot learning performance [4, 5]. Our method is model-agnostic and can be seamlessly integrated into existing imitation-learning or VLA architectures. Therefore, we believe that our method can be used to improve model performance in such large-scale datasets. We will apply our method to train larger VLA models on these larger-scale datasets in our future work, and we are actively pursuing.
>
>
> >**Response for Question 1**: What are the main technical differences between the proposed method and Celik et al., 2022; 2024?
>
> Please see the reply for the first weakness.
>
> [1] Robotwin 2.0: A scalable data generator and benchmark with strong domain randomization for robust bimanual robotic manipulation. arXiv preprint 2025.
>
> [2] Sparse Diffusion Policy: A Sparse, Reusable, and Flexible Policy for Robot Learning. Conference on Robot Learning, 2025.
>
> [3] Efficient Diffusion Transformer Policies with Mixture of Expert Denoisers for Multitask Learning. The Thirteenth International Conference on Learning Representations, 2025.
>
> [4] A careful examination of large behavior models for multitask dexterous manipulation. arXiv preprint 2025.
>
> [5] $\pi_0 $: A Vision-Language-Action Flow Model for General Robot Control. arXiv 2024.

---

### Author Response · Authors · 2025-11-21
**Summary of Changes for Rebuttal**

Dear Reviewers,

We sincerely thank all reviewers for their thoughtful and constructive feedback, which significantly improves the quality of our paper. Based on the comments, we make several substantial revisions and additions:

## **Supplementary Experiments in Simulation**
We conduct additional simulation experiments on the RoboTwin 2.0 [1] benchmark. We compare our method against two relevant baselines based on Diffusion Policy and MoE. A single policy is trained on all eight tasks and evaluated with 100 trials per task. The results show that **Di-BM** consistently outperforms the baselines in multi-task learning:

| Method | Adjust bottle | Beat block hammer | Click alarmclock | Dump bin bigbin | Grab roller | Handover block | Handover mic | Lift pot | Total |
|--------|---------------|-------------------|------------------|------------------|--------------|----------------|--------------|----------|--------|
| DP     | 0.82          | 0.06              | 0.85             | 0.55             | **0.95**     | 0.31           | 0.52         | 0.83     | 0.61   |
| SDP[2] | **0.95**     | 0.15              | 0.73           | 0.35             | 0.79         | 0.05           | 0.42         | 0.24     | 0.46   |
| MoDE[3] | 0.82        | 0.15            | 0.73             | 0.59             | 0.89         | 0.33           | 0.56         | 0.68     | 0.59   |
| Di-BM  | 0.83          | **0.41**          | **0.88**         | **0.60**         | 0.92         | **0.72**       | **0.78**     | **0.91** | **0.76** |

## **Ablation Study on the KL Regularization Coefficient**
We perform an ablation study on the KL regularization coefficient $\beta$ using the RoboTwin 2.0 [1] benchmark. We observe that:

- A **large** value of $\beta$ over-constrains the experts and prevents meaningful specialization, causing the model to behave similarly to a standard diffusion policy.
- A **very small** value of $\beta$ leads experts to “slack off” on more difficult parts of the task, resulting in reduced performance.

These findings support our hypothesis that, with an appropriate choice of $\beta$, each expert is able to specialize in distinct skills.

| $\beta$ | Avg. Success |
|---------|--------------|
| 0.001   | 0.23         |
| 0.01    | **0.76**     |
| 0.1     | 0.59         |
| 1       | 0.60         |
| 10      | 0.63         |

## **Expanded Related Work**
We add citations and discussions of several important related works [2, 3, 4, 5, 6], ensuring a more complete contextualization of our contributions.

## **Clarity Enhancements**
We further refine the paper by providing clearer explanations of the training procedure, dataset details, task difficulty, and computational cost.

We mark changes in the paper in blue. These improvements demonstrate the effectiveness of **Di-BM** as a multi-task learning and robotic manipulation policy, offering relevant and novel contributions to the Robot Learning community.

Sincerely,
The Authors

[1] Robotwin 2.0: A scalable data generator and benchmark with strong domain randomization for robust bimanual robotic manipulation

[2] Sparse diffusion policy: A sparse, reusable, and flexible policy for robot learning

[3] Efficient Diffusion Transformer Policies with Mixture of Expert Denoisers for Multitask Learning.

[4] Mixture-of-experts network with task-oriented perturbation for visual reinforcement learning

[5] Information maximizing curriculum: A curriculum-based approach for learning versatile skills

[6] Curriculum-based imitation of versatile skills

---

### Author Response · Authors · 2025-12-01
**Summary Comment to AC**

Dear Area Chair,

Thank you and all reviewers for your time and the valuable feedback on our submission, *"Learning Diverse Skills for Behavior Models with Mixture of Experts"* (ID: 10592). The reviewers' comments significantly improve the quality of our paper. Based on the provided feedback, we revise our manuscript and offer detailed explanations in our responses to address all raised concerns and questions. Furthermore, several reviewers acknowledge that our responses satisfactorily resolve their concerns and indicate their willingness to increase their scores (**Reviewer myRN** and **Reviewer o12B**). The main revisions, additions, and clarifications made during the rebuttal process are as follows:

1. We clarify the main differences, novelty, and contributions of our method compared with [1], emphasizing that our approach is not a simple adaptation of [1] from the reinforcement learning setting to the imitation learning setting in our responses to the reviewers. We also provide a more detailed explanation in the revised version of the paper. Please refer to **Responses for Weakness 1, Question 1 (Reviewer 43vA)** and **Weakness 1, Question 4 (Reviewer tBc1)**.


2. We add simulation experiments on the RoboTwin 2.0 [2] benchmark and compare our method with related approaches [3,4] to improve reproducibility and to determine the precise contribution and novelty of our work. The experimental results show that our Di-BM consistently outperforms the baselines in multi-task learning. Please refer to **Responses for Weakness 2 (Reviewer 43vA)**, **Weakness 3 (Reviewer jt5F)**, **Weakness 4 (Reviewer myRN)**, and **Weakness 1,3 (Reviewer o12B)**.

3. We add citations and discussions for some related works [3,4,5,6,7] in the revised version according to the reviewers' comments. In our responses to the reviewers, we further clarify the key differences between our method and these approaches, as well as the advantages of our method. Please refer to **Responses for Weakness 1 (Reviewer jt5F)**, **Weakness 1,2,3 (Reviewer myRN)**, and **Weakness 1 (Reviewer o12B)**.



4. We add an ablation study on the KL regularization coefficient $\beta$ according to the reviewers' suggestions and concerns. We also provide explanations regarding the selection of this hyperparameter in our responses to the reviewers. Please refer to **Responses for Weakness 5 (Reviewer myRN)** and **Weakness 2 (Reviewer tBc1)**.

5. For several additional questions and requests raised by the reviewers—such as the detailed descriptions of the dataset, robot type, and task difficulty (**Responses for Weakness 2 (Reviewer jt5F)**), the analysis of computational overhead (**Responses for Weakness 3 (Reviewer tBc1)**), the precise meaning of the 'multi-task training' setting (**Responses for Weakness 2 (Reviewer o12B)**), and the explanation for omitting the entropy bonus in the optimization objective (**Responses for Question 1, 2 (Reviewer o12B)**)—we provide thorough explanations in our responses and include the corresponding clarifications in the revised version of the paper.

We thoroughly respond to all reviewers' comments point by point and implement the corresponding revisions and additions in the manuscript. This includes addressing common concerns raised by multiple reviewers, such as the lack of simulation experiments and missing some related works.

Thank you for your time and consideration.

Best regards,

The Authors of Submission 10592

---

[1] Acquiring diverse skills using curriculum reinforcement learning with mixture of experts. ICML, 2024.
[2] Robotwin 2.0: A scalable data generator and benchmark with strong domain randomization for robust bimanual robotic manipulation. arXiv preprint, 2025.
[3] Sparse diffusion policy: A sparse, reusable, and flexible policy for robot learning. CoRL, 2025.
[4] Efficient Diffusion Transformer Policies with Mixture of Expert Denoisers for Multitask Learning. ICLR, 2025.
[5] Mixture-of-experts network with task-oriented perturbation for visual reinforcement learning. ICML, 2025.
[6] Information Maximizing Curriculum: A Curriculum-Based Approach for Imitating Diverse Skills. NeurIPS, 2023.
[7] Curriculum-based imitation of versatile skills. ICRA, 2023.

---

### Meta-Review · Area_Chair_29ex · 2026-01-08

**Summary:**

The reviewers generally acknowledge the paper’s clear presentation, the efficacy of utilizing MoE with supporting analysis, and the strong results in real-world experiments. However, several consistent concerns emerged, including:
- Limited novelty, particularly due to missing comparisons with relevant baselines and heavy reliance on the work of Di-SkilL (Celik et al. 2024).
- Absence of common simulation experiments and small/insufficient real-world experiments.
- Further clarifications on sensitivity to KL hyperparameter as well as model details and experimental settings.
- Questions about the behavior and underlying mechanisms of the experts.
- Uncertainty about the applicability of the method to e.g., goal-conditioned, score-based policies and large-scale datasets.

**Reviewer Concerns:**

- Simulation results on RoboTwin 2.0 with three baselines (including one MoE method) are included.
- Related work discussion has been expanded.
- New results on KL hyperparameter sensitivity are provided.
- Additional clarifications of e.g., the expert behavior and the computational cost, are provided.

----

- The novelty issues remain. Despite the described differences, the core idea still shares from Di-SkilL. The adaptation choices as well as the original Di-SkilL are not included in experiments, which is insufficient to demonstrate the novelty or superiority of the proposed approach.
- Real-world evaluation remains limited, with only 10 trials per task.
- Applicability to other policies and large-scale datasets is discussed but only mentioned as future work and not empirically or theoretically validated.

**Reviewer Scores:**

43vA is likely to maintain the score. Simulation experiments are added, but no evaluation results support the contributions over Di-SkilL or the effectiveness on large-scale datasets.

jt5F is likely to maintain the score. Similarly, simulation experiments are added, but no evaluation results support the contributions over Di-SkilL or the effectiveness on other types of policies.

myRN mentioned they would increase the score by 2. The added simulation results, KL ablation, and discussion of the related works address most concerns.

tBc1 is likely to maintain the score. Method novelty remains a concern. Although sensitivity to hyperparameter is explored with new experiments, it is still unclear whether the approach works on more complex tasks or datasets.

o12B mentioned to increase the score by 2. The added simulation results and clarifications regarding the experiments and methodology addressed the concerns.

---

### Decision · Program_Chairs · 2026-01-26

Reject